# MoMa: A Simple Modular Learning Framework for Material Property Prediction

**Botian Wang**[1,2*]    **Yawen Ouyang**[1*]    **Yaohui Li**[3*]
**Mianzhi Pan**[3]    **Yuanhang Tang**[1]    **Yiqun Wang**[1]    **Haorui Cui**[2]
**Jianbing Zhang**[3]    **Xiaonan Wang**[4]    **Wei-Ying Ma**[1,5]    **Hao Zhou**[1]

[1] Institute for AI Industry Research (AIR), Tsinghua University
[2] Department of Computer Science and Technology, Tsinghua University
[3] School of Artificial Intelligence, Nanjing University
[4] Department of Chemical Engineering, Tsinghua University
[5] Hong Kong Institute of AI for Science, City University of Hong Kong

## Abstract

Deep learning methods for material property prediction have been widely explored to advance materials discovery. However, the prevailing pre-train paradigm often fails to address the inherent diversity and disparity of material tasks. To overcome these challenges, we introduce MoMa, a simple **Mo**dular framework for **Ma**terials that first trains specialized modules across a wide range of tasks and then adaptively composes synergistic modules tailored to each downstream scenario. Evaluation across 17 datasets demonstrates the superiority of MoMa, with a substantial 14% average improvement over the strongest baseline. Few-shot and module scaling experiments further highlight MoMa's potential for real-world applications. Pioneering a new paradigm of modular material learning, MoMa is open-sourced to foster broader community collaboration.

## 1 Introduction

Accurate and efficient material property prediction is critical for accelerating materials discovery. Key properties such as formation energy and band gap are fundamental in identifying stable and functional materials (Masood et al., 2023; Riebesell et al., 2025). While traditional approaches such as density functional theory offer high precision (Jain et al., 2016), their prohibitive computational cost limits their practicality for large-scale screening (Fiedler et al., 2022; Lan et al., 2023).

Recently, deep learning methods have been developed to expedite traditional approaches (Xie & Grossman, 2018; Griesemer et al., 2023). Pre-trained force field models, in particular, have shown remarkable success in generalizing to a wide spectrum of material property prediction tasks (Shoghi et al., 2024; Rhodes et al., 2025; Wood et al., 2025), outperforming specialized models trained from scratch. These models are typically pre-trained on the potential energy surface (PES) data of materials (Barroso-Luque et al., 2024) and then fine-tuned for the target downstream task.

Despite these advances, we identify two key challenges that undermine the effectiveness of current deep learning models for material property prediction: **diversity** and **disparity**.

First, material tasks exhibit significant diversity (Fig. 1) which challenges the generalizability of existing models. For instance, prevailing force-field models are only trained on PES-derived properties (e.g., force, energy, and stress) mostly focusing on crystalline materials (Yang et al., 2024b; Barroso-Luque et al., 2024). However, material tasks span a much wider variety of systems (e.g., crystals, organic molecules) and properties (e.g., thermal stability, electronic behavior), making it difficult for methods trained on a limited set of data to generalize across the full spectrum of tasks.

Second, the disparate nature of material tasks presents huge obstacles for jointly training a broad span of tasks in one model. Material systems vary significantly in atomic composition, bonding

---

*Equal Contribution. Correspondence to Hao Zhou (zhouhao@air.tsinghua.edu.cn) and Jianbing Zhang (zjb@nju.edu.cn). MoMa is open-sourced at `https://github.com/GenSI-THUAIR/MoMa`.

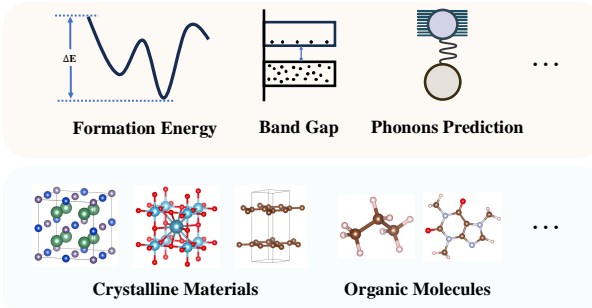 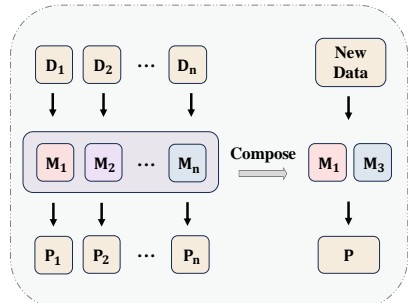

Figure 1: Illustration of the diversity of material properties (top) and systems (down). Material tasks are also disparate, with different laws governing diverse properties and systems. These characteristics pose challenges for material property prediction models.

Figure 2: The modular learning scheme in MoMa trains and stores a broad spectrum of material tasks as modules, and adaptively composes them given a new material property prediction task.

and structural periodicity, while their properties are governed by distinct physical laws. For example, mechanical strength in metals is primarily influenced by atomic bonding and crystal structure, whereas electronic properties like conductivity are determined by the material's electronic structure. Consequently, training a single model across a wide range of tasks (Shoghi et al., 2024) may lead to knowledge conflicts, hindering the model's ability to effectively adapt to downstream scenarios.

Drawing inspiration from modular deep learning (Pfeiffer et al., 2023), we propose MoMa, a **Mo**dular framework for **Ma**terial property prediction. Respecting the **diversity** challenge, MoMa trains multiple high-resource property prediction datasets into transferrable modules to support a wide-span of downstream tasks. In parallel, to address the **disparity** challenge, MoMa encapsulates each task within a specialized module during training to avoid interference. Furthermore, in adapting to each downstream task, MoMa adaptively integrates a synergistic combination of modules to mitigate knowledge conflicts. A high-level abstraction of MoMa is provided in Fig. 2.

Specifically, MoMa comprises two major stages: (1) *Module Training & Centralization*. MoMa trains dedicated modules for a diverse range of material tasks, offering two versions: full modules and memory-efficient adapter modules. These modules are centralized in MoMa Hub, which enables knowledge reuse while keeping proprietary data private. (2) *Adaptive Module Composition (AMC) & Fine-tuning*. We devise AMC, a *representation-driven, training-free* module composition algorithm. Given a target task, AMC first estimates the performance of each module via $k$NN label propagation in representation space. It then infers a weighted module composition by solving a convex optimization problem over a justified proxy error. The composed module is further fine-tuned for improved downstream adaptation. Together, MoMa offers a flexible solution to achieve scalable modular learning for material property prediction.

Empirical results across 17 downstream tasks showcase the superiority of MoMa, outperforming all baselines in **16/17** tasks, with an average improvement of **14%** compared to the best non-modular baseline. In *few-shot* settings, which are common in materials science, MoMa achieves even larger performance gains to the conventional pre-train then fine-tune paradigm. Additionally, MoMa shows improved average improvements as we scale the number of modules in the MoMa Hub, and the AMC-optimized weights provide valuable insights into relationships between material properties. The code and trained modules of MoMa are open-sourced, and we envision MoMa becoming a pivotal platform for the modularization and distribution of materials knowledge, fostering deeper community engagement to accelerate materials discovery.

## 2 RELATED WORK

### 2.1 MATERIAL PROPERTY PREDICTION WITH DEEP LEARNING

Deep learning methods have been widely adopted for predicting material properties (De Breuck et al., 2021). The seminal CGCNN model (Xie & Grossman, 2018) represents crystalline materials

with multi-edge graphs and applies graph neural networks for representation learning. Subsequent work (Choudhary & DeCost, 2021; Das et al., 2023; Yan et al., 2024; Taniai et al., 2024) has focused on improving neural network architectures to better model the inductive biases of crystals.

Another line of work develops pre-training strategies for materials (Jha et al., 2019; Magar et al., 2022; Wang et al., 2025). Recently, a series of large force field models (Merchant et al., 2023; Batatia et al., 2023; Neumann et al., 2024) are trained on massive Potential Energy Surface data (Barroso-Luque et al., 2024) and achieve remarkable accuracy in material tasks (e.g. thermal stability prediction (Riebesell et al., 2025)). Notably, the JMP model (Shoghi et al., 2024), trained across multiple domains (small molecules, catalysts, etc.), performs impressively when fine-tuned on both molecular and crystalline tasks.

Extending beyond these methods, MoMa offers a modular strategy to centralize diverse material knowledge into modules and adaptively compose them, yielding superior downstream performance.

## 2.2 MODULAR DEEP LEARNING

Modular deep learning (Pfeiffer et al., 2023; Xiao et al., 2024) represents a promising paradigm where parameterized modules are composed, selected, and aggregated for function specialization and reuse. Notable examples of modular networks include mixture-of-experts (Jacobs et al., 1991; Shazeer et al., 2016), adapters (Houlsby et al., 2019) and LoRA (Hu et al., 2021). Recently, we have seen increasing applications of modular methods across domains such as NLP (Pfeiffer et al., 2020; Huang et al., 2024; Tan et al., 2024) and CV (Puigcerver et al., 2020; Pham et al., 2024), where its strengths in flexibility and minimizing negative interference have been demonstrated.

An important aspect of modular learning is how modules are weighted prior to composition. Previous adaptive module composition approaches can be broadly grouped into (1) search-based methods that iteratively optimize weights based on downstream predictive performance after composition (Huang et al., 2024; Akiba et al., 2025), and (2) router-based methods that learn composition weights via an additional routing network (Muqeeth et al., 2023; Lu et al., 2024). Crucially, both paradigms rely on the downstream prediction error of the composed model to guide weight allocation. However, this dependence is problematic in material settings: high task disparity makes the error signals (from arbitrary module mixtures) noisy and unstable for search-based methods, while data-scarcity provides insufficient supervision for router learning. Additionally, loading all material modules during router training becomes prohibitively costly as the number of module scales.

In the context of material property prediction, modular learning remains largely under-explored. The most related work is the router-based mixture-of-experts method MoE-(18) (Chang et al., 2022), which loads all available modules and learns a routing network for embedding aggregation.

## 3 PROPOSED FRAMEWORK: MOMA

MoMa is a simple modular framework targeting the diversity and disparity of material property prediction tasks. MoMa involves two major stages. In the first stage (Section 3.1), we train and centralize modules for a diverse range of material systems and properties into MoMa Hub. In the second stage (Section 3.2), we devise a *representation-driven, training-free* algorithm to adaptively select and compose MoMa hub modules for a target task, and then fine-tune the composed model. A visual overview of MoMa is shown in Figure 3.

### 3.1 MODULE TRAINING & CENTRALIZATION

To better exploit the transferrable knowledge of open-source material property prediction datasets, we first train distinctive modules for each high-resource material task, and subsequently centralize these modules to constitute MoMa Hub.

**Module Training** Leveraging the power of state-of-the-art material property prediction models, we choose to employ a pre-trained backbone encoder $f$ as the initialization for training each MoMa module. Note that MoMa is independent of the backbone model choice, which enables smooth integration with other pre-trained backbones.

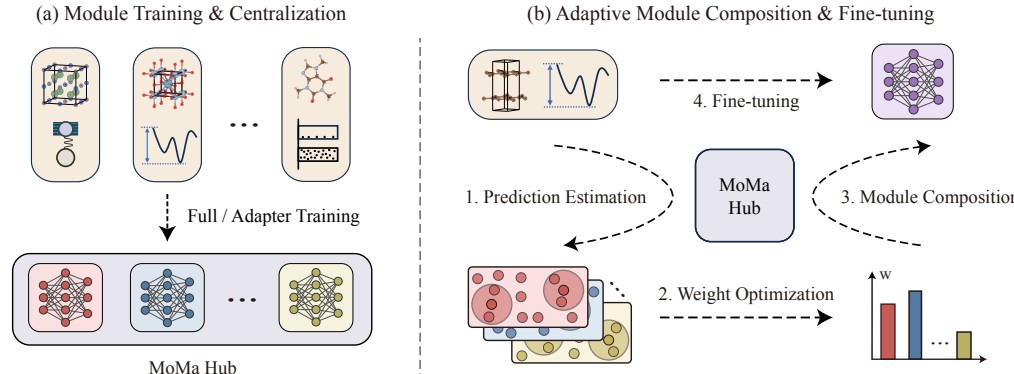

Figure 3: The MoMa framework. (a) During the Module Training & Centralization stage (Section 3.1), MoMa trains full and adapter modules for a wide spectrum of material tasks, constituting the MoMa Hub; (b) The Adaptive Module Composition (AMC) & Fine-tuning stage (Section 3.2) leverages the modules in MoMa Hub to compose a tailored module for each downstream task. The AMC algorithm comprises three steps: 1. Prediction Estimation; 2. Weight Optimization; 3. Module Composition. The composed module is further fine-tuned on the task for better adaptation.

We provide two parametrizations for the MoMa modules: the **full** module and the **adapter** module. For the full module, we directly treat each fully fine-tuned model backbone as a standalone module. The adapter module, in contrast, serves as a parameter-efficient alternative where adapter layers (Houlsby et al., 2019) are inserted between each layer of the backbone. The adapters are updated and the rest of the backbone is frozen. All adapters trained for a given task are collectively treated as one module. This implementation trade-offs the downstream performance for a much lower GPU memory cost during training, making it especially suitable for compute-constrained settings. When training converges, all module parameters are stored into a centralized repository $\mathcal{H}$ termed MoMa Hub, formally:

$$\mathcal{H} = \{g_1, g_2, \ldots, g_N\}, \quad g_i = \begin{cases} \theta_f^i & \text{(full module)} \\ \Delta_f^i & \text{(adapter module)} \end{cases}$$

where $\theta_f^i$ and $\Delta_f^i$ denote the full and adapter module parameters for the $i^{\text{th}}$ task and encoder $f$.

**Module Centralization** To support a wide array of downstream tasks, MoMa Hub needs to include modules trained on diverse material systems and properties. Currently, MoMa Hub encompasses 18 material property prediction tasks selected from the Matminer datasets (Ward et al., 2018) with over 10000 data points. These tasks span across a large range of material properties, including thermal properties (e.g. formation energy), electronic properties (e.g. band gap), mechanical properties (e.g. shear modulus), etc. For more details, please refer to Section C.1. Note that MoMa is designed to be task-agnostic and may readily support a larger spectrum of tasks in the future.

An important benefit of the modular design of MoMa Hub is that it preserves proprietary data, which is prevalent in the field of materials, enabling privacy-aware contribution of new modules. Therefore, MoMa could serve as an open platform for the modularization of materials knowledge.

## 3.2 ADAPTIVE MODULE COMPOSITION & FINE-TUNING

Given a labeled material property prediction dataset $\mathcal{D} = \{(x_i, y_i)\}_{i=1}^M$, the goal of the second stage is to customize a task-specific model by composing modules from MoMa Hub. Due to the diversity and disparity of material tasks, blindly combining modules often leads to suboptimal performance. The composition must be *adaptive*, composing only the most synergistic modules for each task. Furthermore, given the vast and expanding scale of the Hub, the method must be *data-driven and efficient*, avoiding reliance on human expertise or prohibitively expensive exhaustive search.

However, satisfying these requirements is non-trivial for existing adaptive weighting paradigms. As discussed in Section 2.2, both search-based and router-based methods rely on downstream predic-

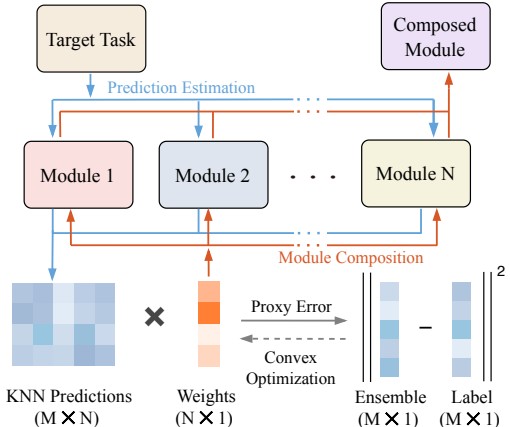

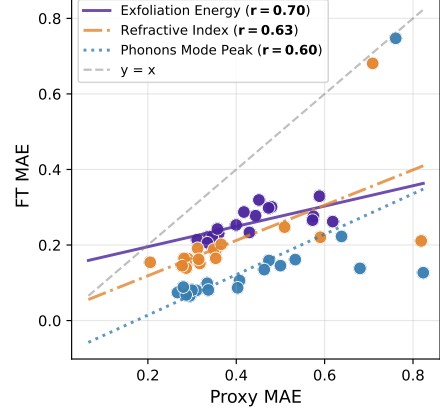

Figure 4: An analytical decomposition of AMC. **Blue** arrows: per-module $k$NN prediction estimation in representation space on target task. **Black** arrows: convex optimization of ensemble proxy error to obtain composition weights. **Orange** arrows: weight-space module composition to construct the final composed module.

Figure 5: Scatter plot showing the relationship between $k$NN proxy MAE and post–fine-tuning MAE of standalone MoMa Hub modules on three representative tasks. Colored lines are linear fits. We observe a clear positive correlation with Pearson's $r > 0.6$.

tion error derived from composed module as the supervision signal. In our setting, this signal is less reliable: the high disparity of modules in inputs (e.g. crystals vs. molecules) and targets (e.g. energies vs. band gaps) induces highly heterogeneous representation spaces. Hence module mixtures yield unstable representations and uninformative error signals, resulting in a noisy optimization landscape that hampers search-based methods. Moreover, the scarcity of downstream data makes router training difficult and prone to overfitting.

To address these limitations, we devise the Adaptive Module Composition (AMC) algorithm. Instead of relying on prediction error supervision, AMC adopts a *representation-driven* and *training-free* strategy. Specifically, it first estimates per-module performance via $k$NN in the representation space, and then solves for optimal composition weights by minimizing an ensemble *proxy error* via convex optimization. This allows AMC to efficiently identify synergistic compositions without iterative search or extra trainable parameters. We now introduce AMC in detail. An analytical figure of AMC is provided in Fig. 4, with its formal formulation in Algorithm 1.

**Representation-driven Prediction Estimation** AMC begins by estimating the affinity of each module to the downstream task. To bypass the unstable optimization landscape of arbitrary module mixtures, we first evaluate the intrinsic representation quality of each module individually. We posit that a task-aligned module should map materials with similar properties to adjacent points in the embedding space.

Formally, for each module $g_j \in \mathcal{H}$, we encode the training data $\mathcal{D}$ into representations $\mathcal{X}^j = \{\mathbf{x}_1^j, \ldots, \mathbf{x}_M^j\}$. We then perform leave-one-out $k$NN label propagation (Iscen et al., 2019) to obtain a prediction $\hat{y}_i^j$ for each instance:

$$\hat{y}_i^j = \sum_{k \in \mathcal{N}_i} \frac{f_d\left(\mathbf{x}_i^j, \mathbf{x}_k^j\right)}{Z_i^j} y_k, \quad Z_i^j = \sum_{k \in \mathcal{N}_i} f_d\left(\mathbf{x}_i^j, \mathbf{x}_k^j\right). \tag{1}$$

where $\mathcal{N}_i$ denotes the indices of the $K$ nearest neighbors of $\mathbf{x}_i^j$ within $\mathcal{X}^j$, and $f_d$ is the exponential cosine similarity function.

We choose $k$NN as the estimator because it directly probes the local geometry of the representation space without introducing learnable parameters. This strictly aligns with our training-free design principle and ensures robustness against overfitting on data-scarce tasks.

**Training-free Module Weight Optimization**    With the module-wise performance estimates $\{\hat{\mathbf{y}}^j\}_{j=1}^N$ from the representation space, our goal is to identify an optimal weight vector $\mathbf{w} \in \mathbb{R}^N$ (where $w_j$ denotes the weight of module $j$) to compose these modules. While the ideal objective is to minimize the validation error of the fine-tuned model, searching this space directly is computationally infeasible due to combinatorial explosion. Instead, inspired by ensemble learning (Zhou et al., 2002; Zhou, 2016), we propose to use the prediction error of the weighted ensemble (prior to fine-tuning) as a *proxy error* to guide weight optimization.

Specifically, we formulate the composition prediction as a weighted sum of the individual module estimations. The proxy error $E_{\mathcal{D}}$ is defined as the mean squared error between the ensemble prediction and the ground truth labels on the training set:

$$E_{\mathcal{D}}(\mathbf{w}) = \frac{1}{M}\Big\| \sum_{j=1}^N w_j \hat{\mathbf{y}}^j - \mathbf{y} \Big\|_2^2. \tag{2}$$

We can further cast Eq. (2) to a constrained optimization problem:

$$\operatorname*{argmin}_{\mathbf{w}} E_{\mathcal{D}}(\mathbf{w}), \quad \text{s.t.} \ \sum_{j=1}^N w_j = 1, \ w_j \geq 0. \tag{3}$$

Since the objective is convex and the constraints define a convex feasible set, the problem admits a global optimum that can be reliably obtained by standard solvers. Moreover, this weight selection is *training-free* since it introduces no additional learnable parameters and requires no gradient-based updates or hyperparameter tuning beyond the optimizer settings.

**Justification for Using the Proxy Error**    A central premise of AMC is that the $k$NN-based proxy error (Eq. 2 is a reliable indicator of the final model performance. Theoretically, we provide a formal risk analysis in Section B to show that, under reasonable assumptions, minimizing this proxy error bounds the risk of the subsequently fine-tuned model. Empirically, when measured in MAE to align with downstream metrics, we observe a strong Pearson correlation ($> 0.6$) between the per-module proxy errors and their post-fine-tuning performance (see Fig. 5 and Section D.1 for detailed discussion). This indicates that the proxy is a reliable predictor of final performance and supports its use for weight optimization.

**Weight-space Module Composition**    Once the optimal weight vector $\mathbf{w}^*$ is obtained, we compose a single customized module $g_{\mathcal{D}}$ for the target task. Inspired by recent advances in model merging (Wortsman et al., 2022; Ilharco et al., 2022; Yu et al., 2024; Yang et al., 2024a), we merge the modules in their weight space: $g_{\mathcal{D}} = \sum_{j=1}^N w_j^* g_j$.

The validity of this averaging is supported by linear mode connectivity (Frankle et al., 2020; Zhou et al., 2023; 2024). Since all modules originate from a common pre-trained initialization, their parameters remain structurally compatible despite task-specific divergence. This ensures that the composed module serves as a stable and well-conditioned initialization for downstream fine-tuning.

**Downstream Fine-tuning**    Finally, to better adapt to the downstream task $\mathcal{D}$, the composed module $g_{\mathcal{D}}$ is appended with a task-specific head and then fine-tuned on $\mathcal{D}$ to convergence.

## 4    EXPERIMENTS

In this section, we conduct comprehensive experiments to demonstrate the empirical effectiveness of MoMa. The experimental setup is outlined in Section 4.1. The main results, discussed in Section 4.2, show that MoMa **substantially outperforms** baseline methods. Additionally, we extend MoMa to **more architectures** in Section 4.3 and conduct an **in depth examination of AMC** in Section 4.4. Confronted with the data scarcity challenge common in real-world materials discovery settings, we evaluate MoMa's few-shot learning ability in Section 4.5, where it achieves **even larger** performance gains compared to baselines. To further highlight the **flexibility and scalability** of MoMa, we extend MoMa Hub to include molecular datasets and present a scaling analysis of MoMa Hub in Section 4.6. Finally, we visualize the module weights optimized by AMC in Section 4.7, highlighting MoMa's potential for providing **valuable insights** into material properties.

Table 1: **Main results for 17 material property prediction tasks.** The best MAE for each task is highlighted in **bold** and the second best result is underlined. The result for each task are the average of five data splits, reported to three significant digits. For each method, the standard deviation of the test MAE across five random seeds is shown in parentheses. Additionally, the average rank and its standard deviation across the 17 datasets are provided to reflect the consistency of each method.

| Datasets | CGCNN | MoE-(18) | UMA | JMP-MT | JMP-FT | MoMa (Adapter) | MoMa (Full) |
|---|---|---|---|---|---|---|---|
| Experimental Band Gap (eV) | 0.471 (0.008) | 0.374 (0.008) | 0.355 (0.037) | 0.377 (0.005) | 0.358 (0.014) | 0.359 (0.009) | **0.305** (0.006) |
| Formation Enthalpy (eV/atom) | 0.193 (0.015) | 0.0949 (0.0016) | 0.192 (0.020) | 0.134 (0.001) | 0.168 (0.007) | 0.158 (0.009) | **0.0839** (0.0013) |
| 2D Dielectric Constant | 2.90 (0.12) | 2.29 (0.01) | 2.34 (0.47) | 2.25 (0.06) | 2.35 (0.07) | 2.31 (0.04) | **1.89** (0.03) |
| 2D Formation Energy (eV/atom) | 0.169 (0.006) | 0.106 (0.005) | 0.120 (0.03) | 0.140 (0.004) | 0.125 (0.006) | 0.112 (0.002) | **0.0495** (0.0015) |
| Exfoliation Energy (meV/atom) | 59.7 (1.5) | 52.5 (0.8) | 44.4 (11.5) | 42.3 (0.5) | **35.4** (2.0) | **35.4** (0.9) | 36.3 (0.2) |
| 2D Band Gap (eV) | 0.686 (0.034) | 0.532 (0.008) | 0.494 (0.061) | 0.546 (0.020) | 0.582 (0.018) | 0.552 (0.014) | **0.375** (0.006) |
| 3D Poly Electronic | 32.5 (1.1) | 27.7 (0.1) | 32.7 (6.0) | 23.9 (0.2) | 23.3 (0.3) | 23.3 (0.2) | **23.0** (0.1) |
| 3D Band Gap (eV) | 0.492 (0.008) | 0.361 (0.003) | 0.268 (0.016) | 0.423 (0.004) | 0.249 (0.001) | 0.245 (0.002) | **0.200** (0.001) |
| Refractive Index | 0.0866 (0.0014) | 0.0785 (0.0004) | 0.0582 (0.0094) | 0.0636 (0.0006) | 0.0555 (0.0027) | 0.0533 (0.0023) | **0.0523** (0.0010) |
| Elastic Anisotropy | 3.65 (0.11) | 3.01 (0.03) | 3.79 (2.48) | 2.53 (0.26) | **2.42** (0.36) | 2.57 (0.61) | 2.86 (0.28) |
| Electronic Dielectric Constant | 0.168 (0.002) | 0.157 (0.015) | 0.116 (0.038) | 0.137 (0.002) | 0.108 (0.002) | 0.106 (0.002) | **0.0885** (0.0048) |
| Dielectric Constant | 0.258 (0.008) | 0.236 (0.002) | 0.183 (0.034) | 0.224 (0.004) | 0.171 (0.002) | 0.168 (0.002) | **0.158** (0.002) |
| Phonons Mode Peak ($cm^{-1}$) | 0.127 (0.004) | 0.0996 (0.0083) | 0.0811 (0.0087) | 0.0859 (0.0006) | 0.0596 (0.0065) | 0.0568 (0.0009) | **0.0484** (0.0026) |
| Poisson Ratio | 0.0326 (0.0001) | 0.0292 (0.0001) | 0.0225 (0.0014) | 0.0297 (0.0003) | 0.0221 (0.0004) | 0.0220 (0.0003) | **0.0204** (0.0002) |
| Poly Electronic | 2.97 (0.10) | 2.61 (0.13) | 2.33 (0.89) | 2.42 (0.03) | 2.11 (0.04) | 2.13 (0.03) | **2.09** (0.03) |
| Poly Total | 6.54 (0.24) | 5.51 (0.04) | 5.61 (1.49) | 5.52 (0.03) | 4.89 (0.06) | 4.89 (0.04) | **4.86** (0.07) |
| Piezoelectric Modulus | 0.232 (0.004) | 0.208 (0.003) | 0.208 (0.027) | 0.199 (0.002) | 0.174 (0.004) | **0.173** (0.003) | 0.174 (0.001) |
| **Average Rank** | 6.88 (0.33) | 4.71 (1.40) | 4.53 (1.42) | 4.53 (1.28) | 3.12 (1.54) | 2.59 (1.12) | **1.35** (0.86) |

## 4.1 SETUP

**Datasets** To better align with real-world material property prediction settings where labels are usually scarce, we conduct experiments on 17 low-data material property prediction tasks from Matminer (Ward et al., 2018) adhering to Chang et al. (2022). This benchmark offers a comprehensive evaluation of model capability on a wide span of properties critical for material discovery. Refer to Section C.1 for more dataset details.

**Implementation Details** For the pre-trained backbone of MoMa, we employ the open-source JMP model (Shoghi et al., 2024) for representing material systems given its superior performance in property prediction tasks across both crystals and molecules. For a rigorous comparison, we present the MAE averaged across the five splits adopted from Chang et al. (2022). Each experiment is repeated with five random seeds, and the reported standard deviation is computed across the seed-level averages. Additional implementation details, including the details of module architecture, the hyper-parameters for MoMa, and the computational cost, are provided in Section C.2.

**Baseline Methods** We compare the performance of MoMa with five baseline methods: CGCNN (Xie & Grossman, 2018), MoE-(18) (Chang et al., 2022), UMA (Wood et al., 2025), JMP-FT, and JMP-MT (Shoghi et al., 2024). CGCNN represents a classical method without pre-training. MoE-(18) trains separate CGCNN models for the upstream tasks of MoMa, then ensembles them as one model in a mixture-of-experts approach for downstream fine-tuning. UMA is a general-purpose atomic foundation model which achieves state-of-the-art performance in canonical benchmarks (Riebesell et al., 2023). We fine-tune the UMA-Medium model on each downstream task. JMP-FT directly fine-tunes the JMP pre-trained checkpoint on the downstream tasks. JMP-MT trains all tasks in MoMa Hub with a multi-task pretraining scheme and then adapts to each downstream dataset with further fine-tuning. More discussions on baselines are included in Section C.3.

## 4.2 MAIN RESULTS

**Performance of MoMa** As shown in Table 1, MoMa (Full) achieves the best performance with the lowest average rank of 1.35 and 14/17 best results. MoMa (Adapter) follows, with an average rank of 2.59. Together, the two variants hold **16/17** best results. They also exhibit the smallest rank deviations, indicating that MoMa consistently delivers reliable performance across tasks. Notably, MoMa (Full) outperforms JMP-FT in 14 tasks, with an impressive average improvement of 14.0%, highlighting the effectiveness of MoMa Hub modules in fostering material property prediction. Moreover, MoMa (Full) surpasses JMP-MT in 16 of 17 tasks with a substantial average margin of 24.8%, underscoring the advantage of MoMa's modular design in mitigating task interference. Further analyses in this section are done with MoMa (Full) due to its superior performance.

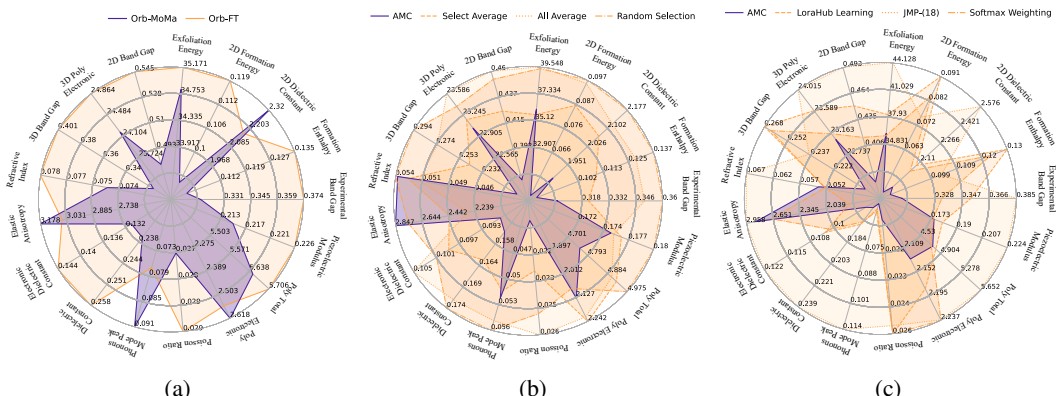

Figure 6: (a) Results with Orb-v2; (b) Ablation study of AMC; (c) Analysis experiments of AMC. The axis represents test set MAE and **smaller area is better**.

**Performance of Baselines**   Among the baseline methods, JMP-FT performs the best with an average rank of 3.12, followed by JMP-MT and UMA with an average rank of 4.53. Though additionally trained on upstream tasks of MoMa Hub, JMP-MT still lags behind JMP-FT. We hypothesize the inherent knowledge conflicts between disparate material tasks pose a tremendous risk to the multi-task learning approach. For UMA, it is primarily pre-trained on force-field datasets as a DFT surrogate, so its inductive bias may transfer less well to non-PES downstream tasks as compared to JMP.

## 4.3 RESULTS WITH MORE ARCHITECTURES

To verify whether MoMa offers consistent benefits in other model backbones beyond JMP, we conduct additional experiments on the GNS architecture (Sanchez-Gonzalez et al., 2020) used by the Orb-v2 model (Neumann et al., 2024), which is not equivariant and much less complex than the GemNet-based architecture (Gasteiger et al., 2022) in JMP. Specifically, we first train and construct an Orb-based MoMa Hub. Then we run AMC and downstream fine-tuning identically as in Section 3.2. The results (Orb-MoMa) are compared with directly fine-tuning the pre-trained Orb model (Orb-FT). The average test MAE are reported on 5 splits and 5 random seeds.

As shown in Fig. 6a, MoMa outperforms in 13/17 tasks and achieves a 6.1% average boost to direct fine-tuning. This indicates that the effectiveness of MoMa is consistent across GemNet-based and GNS-based architectures.

## 4.4 ABLATION & ANALYSIS OF ADAPTIVE MODULE COMPOSITION

**Ablation Study**   We conduct a fine-grained ablation study of AMC with three variants : (1) *Select Average*, which retains the AMC-selected modules (nonzero weights) but averages them uniformly; (2) *All Average*, simply averages all modules in MoMa Hub, which is equivalent to applying the classical Model Soup strategy (Wortsman et al., 2022); (3) *Random Selection*, which picks a random set of modules in MoMa Hub with the same module number as AMC. A visualization of the ablation results is presented in Fig. 6b. The three variants are inferior to AMC in 13, 15 and 15 out of 17 tasks, with an average test MAE increase of 11.0%, 18.0% and 20.2%, respectively. This highlight the effectiveness of both module selection and weighted composition in AMC. The average test MAE of 5 splits are reported on one random seed (hereafter). Furthermore, we show in Section D.2 that AMC is robust to $k$NN configurations and solver tolerances, yielding highly stable weights and consistent post–fine-tuning MAE.

**Analysis Experiments**   To empirically validate the benefit of AMC's representation-driven and training-free pipeline, we replace AMC with three alternatives: (1) *LoRAHub Learning* (Huang et al., 2024), a black-box optimization approach for module composition; (2) *JMP-(18)*, where we train a routing network over the 18 JMP MoMa modules; and (3) *Softmax Weighting*, a non-optimized heuristic based on $k$NN proxy. As shown in Fig. 6c, AMC consistently outperforms all baselines, surpassing the three variants on 15, 17, and 12 tasks with average MAE reductions of

21.8%, 15.5%, and 13.7%, respectively. This shows the benefit of the AMC over search-based, router-based and performance-based alternatives. See more details and discussion in Section C.5.

**Efficiency Analysis** We highlight that AMC is highly efficient: it requires only a single round of forward embedding generation, followed by lightweight $k$NN prediction and convex optimization. For the largest dataset, AMC converges in under 30 seconds. This efficiency enables MoMa to scale to a larger number of modules in future applications. See Section D.3 for a detailed analysis.

## 4.5 PERFORMANCE IN FEW-SHOT SETTINGS

**Motivation & Setup** To better assess the performance of MoMa in real-world scenarios, where labeled material candidates are costly and often scarce (Abed et al., 2024), we construct a few-shot learning setting and compare MoMa with JMP-FT. For each downstream task, we down-sample the training data and apply AMC to compose modules from MoMa Hub, followed by fine-tuning on the sampled subset. The validation and test sets remain consistent with those in the standard setting for robust evaluation. Experiments are conducted under 10-shot and 100-shot conditions, representing few-shot and extremely few-shot scenarios.

Table 2: **Few-shot evaluation.** The average normalized test MAEs of MoMa and JMP-FT under varying data settings. MoMa consistently outperforms JMP-FT in all settings.

|  | 10-shot | 100-shot | Full data |
|---|---|---|---|
| JMP-FT | 0.7003 | 0.4076 | 0.2217 |
| MoMa | **0.5503** | **0.2990** | **0.1871** |

**Results** The average normalized test MAEs[1] for the 17 downstream tasks of MoMa compared to JMP-FT across the full-data, 100-data, and 10-data settings are presented in Table 10. As expected, the test loss increases as the data size decreases, while MoMa consistently outperforms JMP-FT in all settings. Notably, the performance advantage of MoMa is more pronounced in the few-shot settings, with the normalized loss margin widening from 0.03 in the full-data setting to 0.11 and 0.15 in the 100-data and 10-data setting. This suggests that MoMa may offer even greater performance gains in real-world scenarios, where property labels are often limited, thereby hindering the effective fine-tuning of large pre-trained models. Complete results are shown in Section D.4.

## 4.6 SCALING ANALYSIS OF MOMA HUB MODULES

**Motivation & Setup** In this section, we study the scaling behavior of MoMa to understand whether it benefits from a larger MoMa Hub. We first do a hub-scale ablation to progressively expand MoMa hub from 5 to 10 and 18 modules. Then we further expand MoMa Hub to include 12 QM9 modules (Ramakrishnan et al., 2014), which are trained on 12 quantum chemical properties for 134,000 stable small organic molecules. We consistently perform AMC and evaluation across all Hub variants. The full setup is described in Section C.6.

**Results** As presented in Table 3, as the MoMa hub scales, the average normalized test MAE across 17 tasks decreases monotonically (from 0.2040 with 5 modules to 0.1759 with 30 modules), showing no sign of saturation in this regime. The complete results are provided in Table 11 (Section D.5).

To further analyze the effect of adding the 12 QM9 modules, we plot the test-MAE reduction rate against the AMC proxy-error decrease in Fig. 7 for datasets where QM9 modules are selected. We observe that: (1) The integration of QM9 modules leads to an average of 1.7% decrease in test set MAE; (2) a larger reduction in

Table 3: **Scaling with hub size.** Average normalized test MAE decreases as the number of modules in MoMa Hub increases.

| # Modules | 5 | 10 | 18 | 30 |
|---|---|---|---|---|
| Norm. MAE | 0.2040 | 0.1910 | 0.1853 | 0.1759 |

the AMC-optimized proxy error correlates with greater performance improvements post-fine-tuning (Pearson correlation = 0.69). We highlight the task of MP Phonons prediction, which marks a 11.8% decrease in test set MAE after the inclusion of QM9 modules. Overall, these results support our vision of MoMa as a flexible community-driven platform: as more modules are added, downstream performance improves and AMC remains effective at larger scale.

---

[1]Computed by dividing the test MAE of each task by its standard deviation.

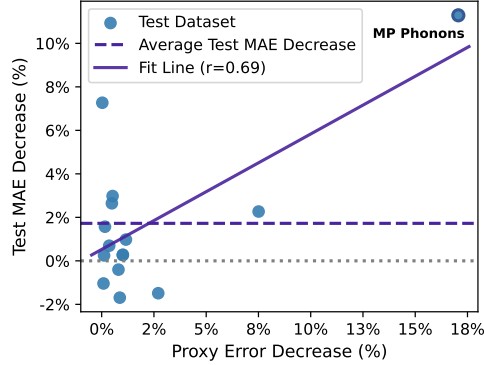

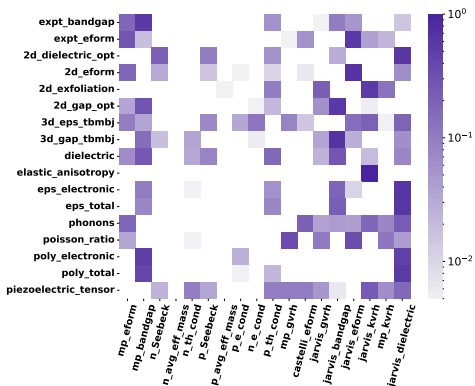

Figure 7: Scatter plot showing the relationship between the test MAE decrease and the proxy error (Eq. (3)) decrease after adding QM9 modules. The solid line represents a linear regression fit, yielding a Pearson correlation of 0.69.

Figure 8: Heat map of AMC weights on one data split. The x-axis represents the task names of the MoMa Hub modules, while the y-axis shows the 17 material tasks in Table 1. Darker color indicates a larger weight.

### 4.7 MATERIALS INSIGHTS MINING

**Motivation**   We argue that the AMC weights derived in Eq. (3) can provide valuable insights into the relationships of material properties. To explore this, we interpret the weights as indicators for the relationships between MoMa Hub modules and downstream tasks. Following Chang et al. (2022), we present a log-normalized visualization of these weights in Fig. 8.

**Results**   We highlight several noteworthy observations: (1) **The weights assigned by AMC effectively capture physically intuitive relationships between material properties.** For instance, in predicting electronic dielectric constants, MoMa assigns high weights to the band gap modules, which is reasonable given the inverse relationship between the dielectric constant and the square of the band gap; (2) **Less intuitive relationships also emerge.** For the task of experimental band gap prediction (row 1), the DFT-computed formation energy module (column 1) is assigned the second-highest weight. In the prediction of dielectric constant (row 9), modules related to thermoelectric and thermal properties (columns 5 and 6) are heavily weighted. However, the first-principles relationship between these tasks is indirect. We hypothesize that in addition to task relevance, other factors such as data distribution and size may also influence the weight assignments for AMC.

## 5 CONCLUSION

In this paper, we present MoMa, a simple modular learning framework for material property prediction. Motivated by the challenges of diversity and disparity in materials, MoMa first trains specialized modules across a wide spectrum of material tasks, constituting MoMa Hub. We then introduce the Adaptive Module Composition algorithm, which facilitates tailored adaptation from MoMa Hub to each downstream task by adaptively composing synergistic modules. Experimental results across 17 datasets demonstrate the superiority of MoMa, with few-shot and hub-scaling experiments further highlighting its data efficiency and scalability.

**Limitations and Future Work.** The current scope of our study is limited to crystalline and organic materials. Future work includes expanding MoMa Hub with modules for a wider range of material data and prediction tasks, and examining how MoMa scales with hundreds or thousands of modules, which may yield deeper insights into the modularity of materials knowledge.

**Broader Impact.** As an open-source platform for modularizing and distributing materials knowledge, MoMa enables secure sharing of modules without exposing proprietary data, efficient customization for downstream tasks, and improved prediction accuracy even in low-data scenarios. We envision MoMa fostering a new paradigm of modular material learning and driving broader community collaboration toward accelerated materials discovery.

ACKNOWLEDGMENTS

The authors would thank Junwei Yang, Ziyao Cao, Fanyou Meng, and Yuxuan Song for their valuable feedback on the paper. We also thank the anonymous reviewers for reviewing the draft. This work is supported by the National Science and Technology Major Project (2022ZD0117502), the Natural Science Foundation of China (Grant No. 62376133, 62406170), and Wuxi Research Institute of Applied Technologies, Tsinghua University under Grant 20242001120.

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

## A  ALGORITHM FOR ADAPTIVE MODULE COMPOSITION

The formal description of the Adaptive Module Composition algorithm is included in Algorithm 1.

---

**Algorithm 1** Adaptive Module Composition (AMC)

---

1: **Input:** MoMa Hub $\mathcal{H} = \{g_j\}_{j=1}^N$, Downstream training set $\mathcal{D} = \{(X_i, y_i)\}_{i=1}^m$.
2: **Output:** Composed module $g_{\mathcal{D}}$.
3: $\{1.\ Module\ Prediction\ Estimation\}$
4: **for** $j = 1 \to N$ **do**
5:     Generate embeddings $\mathcal{X}^j \leftarrow \{g_j(X_i) \mid (X_i, y_i) \in \mathcal{D}\}$.
6:     Compute prediction vector $\hat{\mathbf{Y}}^j = (\hat{y}_1^j, \ldots, \hat{y}_m^j)$ via leave-one-out $k$-Nearest Neighbors.
7: **end for**
8: $\{2.\ Module\ Weight\ Optimization\}$
9: Let $\mathbf{Y} = (y_1, \ldots, y_m)$ be the vector of true labels from $\mathcal{D}$.
10: Find optimal weights $\boldsymbol{w}^* = (w_1^*, \ldots, w_N^*)$ by solving the convex optimization problem:
11: $\boldsymbol{w}^* \leftarrow \arg\min_{\boldsymbol{w}} \left\| \sum_{j=1}^N w_j \hat{\mathbf{Y}}^j - \mathbf{Y} \right\|_2^2$
12:     **subject to:** $\sum_{j=1}^N w_j = 1$ and $w_j \geq 0$ for all $j \in \{1, \ldots, N\}$.
13: $\{3.\ Module\ Composition\}$
14: $g_{\mathcal{D}} \leftarrow \sum_{j=1}^N w_j^* g_j$
15: **Return** $g_{\mathcal{D}}$

---

## B  THEORETICAL JUSTIFICATION AND ERROR ANALYSIS FOR AMC

In this section, we provide a formal analysis of the $k$NN-based proxy error used in AMC. Specifically, we show that the $k$NN proxy risk $R_{proxy}(w)$ serves as an upper bound for the fine-tuning risk $R_{FT}(w)$ (subject to approximation errors). Consequently, minimizing the empirical approximation of $R_{proxy}(w)$ tightens this bound, thereby providing theoretical justification for using the proxy risk to control the risk of the subsequently fine-tuned model.

### B.1  DEFINITIONS

Let $\theta_i$ denote the parameters of the $i$-th module and define its representation of input $x$ as $g_i(x) := g(\theta_i; x)$. Given weights $w = (w_1, \ldots, w_N) \in \Delta_{N-1}$, define the merged module in parameter space and its representation by

$$\theta_w := \sum_{i=1}^N w_i \theta_i, \qquad g_w(x) := g(\theta_w; x).$$

Let the Bayes regressors associated with each representation be
$$m_i(x) := \mathbb{E}[Y \mid g_i(X) = g_i(x)], \quad m_w(x) := \mathbb{E}[Y \mid g_w(X) = g_w(x)].$$

We define the following risk terms:

- Representation Bayes Risk: $R^*(w) := \mathbb{E}\big[(m_w(X) - Y)^2\big]$

- Fine-tuning Risk: $R_{\mathrm{FT}}(w) := \mathbb{E}\big[(\hat{y}_{\mathrm{FT}}(X; w) - Y)^2\big]$

- Proxy Risk using kNN: $R_{\mathrm{proxy}}(w) := \mathbb{E}\big[(\hat{y}_{\mathrm{proxy}}(X; w) - Y)^2\big]$

- Bayes Ensemble Risk: $R_{\mathrm{ens}}(w) := \mathbb{E}\big[(m_{\mathrm{ens}}(X; w) - Y)^2\big]$, where $m_{\mathrm{ens}}(x; w) := \sum_{i=1}^N w_i m_i(x)$.

**Remark (distribution vs empirical proxy objective).** Practically AMC optimizes the empirical proxy error

$$E_{\mathcal{D}}(w) = \frac{1}{m} \sum_{k=1}^m \Big( \sum_{i=1}^N w_i \hat{y}_{i,k} - y_k \Big)^2,$$

while analysis here is stated for the distribution-level proxy risk $R_{\mathrm{proxy}}(w)$. Under standard generalization results for squared-loss regression and mild capacity control on the family $\{\hat{y}_{\mathrm{proxy}}(\cdot; w) : w \in \Delta_{N-1}\}$, the empirical proxy error $E_{\mathcal{D}}(w)$ concentrates around $R_{\mathrm{proxy}}(w)$; hence we treat $E_{\mathcal{D}}(w)$ in practice as a finite-sample approximation of $R_{\mathrm{proxy}}(w)$.

### B.2 PRELIMINARIES AND ASSUMPTIONS

We recall a standard non-parametric regression result on the consistency of kNN estimators.

**Lemma 1** (Universal kNN $L_2$-Consistency (Stone, 1977; Devroye et al., 1994)). *For each upstream task $i$, consider a training dataset of size $n$, and let $\hat{y}_n^{(i)}$ be the k-nearest neighbor regressor in the feature space $g_i(x)$, where the number of neighbours $k = k_n$ satisfies $k_n \to \infty, \quad k_n/n \to 0$. Define the $L_2$-estimation error*

$$\varepsilon_{\mathrm{kNN}}^{(i)}(n) := \mathbb{E}\big[(\hat{y}_n^{(i)}(X) - m_i(X))^2\big].$$

*Then the kNN regression estimator satisfies the exact risk decomposition*

$$\mathbb{E}\big[(\hat{y}_n^{(i)}(X) - Y)^2\big] = \mathbb{E}\big[(m_i(X) - Y)^2\big] + \varepsilon_{\mathrm{kNN}}^{(i)}(n),$$

*and moreover $\varepsilon_{\mathrm{kNN}}^{(i)}(n) \to 0$ as $n \to \infty$.*

**Remark on Lemma 1.** Our implementation of AMC uses a finite-$K$ kNN ensemble with cosine similarity in the learned representation space. This differs from the asymptotic setting of Lemma 1 (which requires $k_n \to \infty$ and $k_n/n \to 0$), but it serves as a computationally efficient finite-sample approximation. Lemma 1 is used as a *conceptual* tool to justify the kNN-based proxy objective; we do not claim a full universal consistency result for our exact finite-sample variant.

**Assumption 1 (Fine-tuning stability).** There exists $\varepsilon_{\mathrm{opt}} \geq 0$ such that for all $w \in \Delta_{N-1}$,

$$R_{\mathrm{FT}}(w) \leq R^*(w) + \varepsilon_{\mathrm{opt}}.$$

That is, after fine-tuning on top of $g_w$, the resulting predictor is within $\varepsilon_{\mathrm{opt}}$ of the optimal predictor defined on the same representation (McNamara & Balcan, 2017; Mehra et al., 2024).

**Assumption 2 (Representation closeness).** There exists $\delta > 0$ and a high-probability subset $\mathcal{X}_0 \subseteq \mathcal{X}$ such that for all modules $i$ and all $x \in \mathcal{X}_0$,

$$\|g_i(x) - g_w(x)\| \leq \delta.$$

This models the empirical observation that independently fine-tuned modules from the same pre-trained model can often be aligned into a shared, approximately convex basin in parameter space, leading to similar internal representations. Recent work on mode connectivity (Frankle et al., 2020; Entezari et al.) and cross-task linearity (Zhou et al., 2024) supports this assumption, which we adopt here as a structural modeling assumption rather than a general theorem.

**Assumption 3 (stability of latent regressor).** Let $g$ and $g'$ denote any two modules among $\{g_i : i \in [N]\}$ and the merged module $g_w$. Let their associated Bayes regressors be

$$m(x) := \mathbb{E}[Y \mid g(X) = g(x)], \qquad m'(x) := \mathbb{E}[Y \mid g'(X) = g'(x)].$$

There exists $L > 0$ and a high-probability subset $\mathcal{X}_0 \subseteq \mathcal{X}$ such that if

$$\|g(x) - g'(x)\| \leq \delta \qquad \forall x \in \mathcal{X}_0,$$

then the regressors are close in expectation:

$$\mathbb{E}\big[|m(X) - m'(X)|\,\mathbf{1}\{X \in \mathcal{X}_0\}\big] \leq L\delta.$$

We also assume $|Y| \leq B$ almost surely.

**Remark on Assumption 3.** Assumption 3 formalizes a semantic smoothness condition in the learned representation: inputs that are mapped to nearby latent points (i.e., with close $g(x)$ and $g'(x)$) are required to have similar predictive behavior for $Y$ on average. This can be viewed as a Lipschitz-type regularity assumption on the Bayes regressors in representation space, and is consistent with the common inductive bias in deep learning that well-trained encoders should support target functions which do not change abruptly under small perturbations of the latent features.

### B.3 RISK TRANSFER ANALYSIS

**Step 1: kNN Ensemble Approximates the Bayes Ensemble.**   We compare the proxy risk to the Bayes ensemble risk:

$$
\begin{aligned}
\big| R_{\text{proxy}}(w) - R_{\text{ens}}(w) \big| &= \big| \mathbb{E}\big[ (\hat{y}_{\text{proxy}} - Y)^2 - (m_{\text{ens}} - Y)^2 \big] \big| \\
&= \big| \mathbb{E}\big[ (\hat{y}_{\text{proxy}} - m_{\text{ens}})(\hat{y}_{\text{proxy}} + m_{\text{ens}} - 2Y) \big] \big| \\
&\leq 4B\,\mathbb{E}\big[ |\hat{y}_{\text{proxy}} - m_{\text{ens}}| \big] \qquad \text{(by bounded label in Assumption 3)}
\end{aligned}
$$

Next,

$$
\begin{aligned}
\mathbb{E}\big[ |\hat{y}_{\text{proxy}} - m_{\text{ens}}| \big] &= \mathbb{E}\Big[ \Big| \sum_{i=1}^{N} w_i \hat{y}^{(i)} - \sum_{i=1}^{N} w_i m_i \Big| \Big] \\
&\leq \sum_{i=1}^{N} w_i\,\mathbb{E}\big[ |\hat{y}^{(i)} - m_i| \big] \\
&\leq \sum_{i=1}^{N} w_i \sqrt{\mathbb{E}\big[ (\hat{y}^{(i)} - m_i)^2 \big]} \\
&= \sum_{i=1}^{N} w_i \sqrt{\varepsilon_{\text{kNN}}^{(i)}(n)} \qquad \text{(by Assumption 1)} \\
&\leq \sqrt{\sum_{i=1}^{N} w_i\,\varepsilon_{\text{kNN}}^{(i)}(n)} \qquad \text{(by Jensen's inequality)}
\end{aligned}
$$

Define the weighted kNN error

$$
\bar{\varepsilon}_{\text{kNN}}(n; w) := \sum_{i=1}^{N} w_i\,\varepsilon_{\text{kNN}}^{(i)}(n), \qquad \epsilon_1(n; w) := 4B \sqrt{\bar{\varepsilon}_{\text{kNN}}(n; w)}\,.
$$

Then we obtain

$$
\big| R_{\text{proxy}}(w) - R_{\text{ens}}(w) \big| \leq \epsilon_1(n; w).
$$

**Step 2: Bayes Ensemble Approximates the Merged Bayes Predictor.**   We compare $m_{\text{ens}}(x; w)$ and $m_w(x)$ using the triangle inequality,

$$
\begin{aligned}
\mathbb{E}\big[ |m_{\text{ens}}(X; w) - m_w(X)|\, \mathbf{1}\{X \in \mathcal{X}_0\} \big] &= \mathbb{E}\Big[ \Big| \sum_{i=1}^{N} w_i m_i(X) - m_w(X) \Big|\, \mathbf{1}\{X \in \mathcal{X}_0\} \Big] \\
&\leq \sum_{i=1}^{N} w_i\,\mathbb{E}\big[ |m_i(X) - m_w(X)|\, \mathbf{1}\{X \in \mathcal{X}_0\} \big] \\
&\leq L\,\delta \qquad \text{(by Assumption 2 and 3)}
\end{aligned}
$$

Then

$$
\begin{aligned}
\big| R_{\text{ens}}(w) - R^*(w) \big| &= \big| \mathbb{E}\big[ (m_{\text{ens}}(X; w) - Y)^2 - (m_w(X) - Y)^2 \big] \big| \\
&= \big| \mathbb{E}\big[ (m_{\text{ens}}(X; w) - m_w(X))(m_{\text{ens}}(X; w) + m_w(X) - 2Y) \big] \big| \\
&\leq 4B\,\mathbb{E}\big[ |m_{\text{ens}}(X; w) - m_w(X)| \big] \\
&\leq 4B\,L\,\delta + (\text{small error term on } \mathcal{X}_0^c).
\end{aligned}
$$

Absorbing the small-probability contribution from $\mathcal{X}_0^c$ into the constant, we obtain

$$
\big| R_{\text{ens}}(w) - R^*(w) \big| \leq C\,\delta,
$$

where $C := 4BL$.

## B.4 Main Transfer Bound and Guarantee

**Proposition 1.** *Under Assumptions 1–3 and Lemma 1, for any $w \in \Delta_{N-1}$,*

$$R_{\mathrm{FT}}(w) \leq R_{\mathrm{proxy}}(w) + C\,\delta + \varepsilon_{\mathrm{opt}} + \epsilon_1(n; w),$$

*where $C = 4BL$ and $\epsilon_1(n; w) = 4B\sqrt{\bar{\varepsilon}_{\mathrm{kNN}}(n; w)}$.*

*Proof.* From Step 1 we have

$$R_{\mathrm{proxy}}(w) \geq R_{\mathrm{ens}}(w) - \epsilon_1(n; w),$$

and from Step 2

$$R_{\mathrm{ens}}(w) \geq R^*(w) - C\,\delta.$$

Hence

$$R^*(w) \leq R_{\mathrm{proxy}}(w) + C\,\delta + \epsilon_1(n; w).$$

Combining with Assumption 1 yields

$$R_{\mathrm{FT}}(w) \leq R^*(w) + \varepsilon_{\mathrm{opt}} \leq R_{\mathrm{proxy}}(w) + C\,\delta + \varepsilon_{\mathrm{opt}} + \epsilon_1(n; w).$$

Combining this result with the optimality of $\hat{w}$ for the proxy objective yields the following theorem.

**Theorem 1.** *Let $\hat{w} \in \arg\min_{w \in \Delta_{N-1}} R_{\mathrm{proxy}}(w)$ be a minimizer of the proxy risk. Under Assumptions 1–3 and Lemma 1, the fine-tuning risk of the merged encoder with weights $\hat{w}$ satisfies*

$$R_{\mathrm{FT}}(\hat{w}) \leq R_{\mathrm{proxy}}(\hat{w}) + C\,\delta + \varepsilon_{\mathrm{opt}} + \epsilon_1(n; \hat{w}),$$

*where $C = 4B\,L$ and $\epsilon_1(n; \hat{w}) = 4B\sqrt{\sum_{i=1}^{N} \hat{w}_i\, \varepsilon_{\mathrm{kNN}}^{(i)}(n)}$.*

## C Experimental Details

Here we provide more experimental details regarding the datasets, baselines, and implementation.

### C.1 Dataset Details

We primarily adopt the dataset setup proposed by Chang et al. (2022). Specifically, we select 35 datasets from Matminer (Ward et al., 2018) for our study, categorizing them into 18 high-resource material datasets, with sample sizes ranging from 10,000 to 132,000 (an average of 35,000 samples), and 17 low-data datasets, with sample sizes ranging from 522 to 8,043 (an average of 2,111 samples).

The high-resource datasets are utilized for training the MoMa Hub modules, as their larger data volumes are likely to encompass a wealth of transferrable material knowledge. A detailed introduction of these MoMa Hub datasets is included in Table 4.

The low-data datasets serve as downstream tasks to evaluate the effectiveness of MoMa and its baselines. A detailed introduction is included in Table 5. This setup mimics real-world materials discovery scenarios, where downstream data are often scarce. Compared to the benchmark in Chang et al. (2022), we exclude two low-data datasets with exceptionally small data sizes (fewer than 20 test samples) from our experiments, as their limited data could lead to unreliable conclusions.

Following Chang et al. (2022), all datasets are split into training, validation, and test sets with a ratio of 7:1.5:1.5. For the downstream low-data datasets, we follow the exact splitting provided by Chang et al. (2022) to ensure a fair comparison.

### C.2 Implementation Details of MoMa

**Module Architecture Details** We now introduce the architectural details of MoMa modules. Across all our experiments in the main text, the JMP (Shoghi et al., 2024) backbone is adopted due to its comprehensive strength across a wide range of molecular and crystal tasks. JMP is pre-trained on $\sim$ 120 million DFT-generated force-field data across large-scale datasets on catalyst and small molecules. It is a 6-layer GNN model with around 160M parameters which is based on the

Table 4: Datasets for training MoMa Hub modules. **Num** stands for the number of samples in each dataset.

| Datasets | Num | Description |
|---|---|---|
| MP $E_f$ | 132752 | The energy change during the formation of a compound from its elements. Data from Jain et al. (2013). |
| MP $E_g$ | 106113 | The PBE band gaps, calculated using the Perdew-Burke-Ernzerhof (PBE) functional, represent the energy difference between the valence and conduction bands in a material. Data from Jain et al. (2013). |
| MP $G_{VRH}$ | 10987 | VRH-average shear modulus, an approximate value obtained by averaging the shear modulus of polycrystalline materials. Data from Jain et al. (2013). |
| MP $K_{VRH}$ | 10987 | VRH-average bulk modulus, calculated by averaging the Voigt (upper bound) and Reuss (lower bound) bulk moduli. Data from Jain et al. (2013). |
| n-type $\sigma_e$ | 37390 | n-type $\sigma_e$ measures the material's conductivity performance when electrons are the primary charge carriers. Data from Ricci et al. (2017). |
| p-type $\sigma_e$ | 37390 | Similar to n-type $\sigma_e$, with holes as carriers. Data from Ricci et al. (2017). |
| n-type $\kappa_e$ | 37390 | n-type $\kappa_e$ evaluates the efficiency of n-type materials that can conduct both electricity and heat, which is crucial for understanding its performance in thermoelectric applications. Data from Ricci et al. (2017). |
| p-type $\kappa_e$ | 37390 | Similar to n-type $\kappa_e$, with holes as carriers. Data from Ricci et al. (2017). |
| n-type $S$ | 37390 | n-type $S$ denotes the average conductivity eigenvalue, which measures thermoelectric conversion efficiency in the hole-conducting state when electrons act as the primary charge carriers. Data from Ricci et al. (2017). |
| p-type $S$ | 37390 | Similar to n-type $S$, with holes as carriers. Data from Ricci et al. (2017). |
| n-type $\overline{m}_e^*$ | 21037 | n-type $\overline{m}_e^*$ denotes the average eigenvalue of conductivity effective mass, which measures the impact of the electron's effective mass on the electrical conductivity. Data from Ricci et al. (2017). |
| p-type $\overline{m}_e^*$ | 20270 | Similar to n-type $\overline{m}_e^*$, with holes as carriers. Data from Ricci et al. (2017). |
| Perovskite $E_f$ | 18928 | Perovskite $E_f$ refers to the heat of formation of perovskite, the amount of heat released or absorbed when the perovskite structure is formed from its constituent elements. Data from Castelli et al. (2012). |
| JARVIS $E_f$ | 25923 | Formation energy from the JARVIS dataset (Choudhary et al., 2020). |
| JARVIS dielectric constant (Opt) | 19027 | Dielectric constant measures the material's ability to polarize in response to an electric field in two-dimensional systems. Data from Choudhary et al. (2020). |
| JARVIS $E_g$ | 23455 | PBE band gaps from the JARVIS dataset (Choudhary et al., 2020). |
| JARVIS $G_{VRH}$ | 10855 | VRH-average shear modulus from the JARVIS dataset (Choudhary et al., 2020). |
| JARVIS $K_{VRH}$ | 11028 | VRH-average bulk modulus from the JARVIS dataset (Choudhary et al., 2020). |

GemNet-OC architecture (Gasteiger et al., 2022). Note that MoMa is backbone-agnostic and we include results with the Orb model (Neumann et al., 2024) in Section 4.3.

For the full module parametrization, we exclude the output layer and treat the entire GNN backbone as a single module. For the adapter components, we follow the standard implementation of adapter layers (Houlsby et al., 2019). Specifically, an adapter layer is inserted between every two layers of the JMP backbone. Each adapter consists of a downward projection to a bottleneck dimension, followed by an upward projection back to the original dimension. We adopt BERT-style initialization (Devlin, 2018), with the bottleneck dimension set to half of the input embedding dimension. Note that the merging process for adapters is performed in a layer-wise manner. For each backbone layer containing adapters, we compute a weighted average of the parameters from all selected adapter modules. A single scalar weight for each module, determined by AMC, is applied uniformly across all adapter layers belonging to that module.

**Hyper-parameters** For the training of JMP backbone, we mainly follow the hyper-parameter configurations in Shoghi et al. (2024), with slight modifications to the learning rate and batch size. During the module training stage of MoMa, we use a batch size of 64 and a learning rate of 5e-4 for 80 epochs. During downstream fine-tuning, we adopt a batch size of 32 and a learning rate of 8e-5. We set the training epoch as 60, with an early stopping patience of 10 epochs to prevent over-fitting. We

Table 5: Downstream evaluation datasets.

| Datasets | Num | Description |
|---|---|---|
| Experimental Band Gap (eV) | 2481 | The band gap of a material as measured through physical experiments. Data from Ward et al. (2018). |
| Formation Enthalpy (eV/atom) | 1709 | The energy change for forming a compound from its elements, crucial for defining Gibbs energy of formation. Data from Wang et al. (2021); Kim et al. (2017). |
| 2D Dielectric Constant | 522 | The dielectric constant of 2D materials from Choudhary et al. (2017). |
| 2D Formation Energy (eV/atom) | 633 | The energy change associated with the formation of 2D materials from their constituent elements. Data from Choudhary et al. (2017). |
| Exfoliation Energy (meV/atom) | 636 | The energy required to separate a single or few layers from bulk materials. Data from Choudhary et al. (2017). |
| 2D Band Gap (eV) | 522 | The band gap of 2D materials from Choudhary et al. (2017). |
| 3D Poly Electronic | 8043 | Poly electronic of 3D materials from Choudhary et al. (2018). |
| 3D Band Gap (eV) | 7348 | The band gap of 3D materials from Choudhary et al. (2018). |
| Refractive Index | 4764 | The quantitative change of the speed of light as it passes through different media. Data from Dunn et al. (2020); Petousis et al. (2017). |
| Elastic Anisotropy | 1181 | The directional dependence of a material's elastic properties. Data from De Jong et al. (2015a). |
| Electronic Dielectric Constant | 1296 | Electronic dielectric constant refers to the dielectric response caused by electronic polarization under an applied electric field. Data from Petretto et al. (2018). |
| Dielectric Constant | 1296 | Dielectric constant of materials from Petretto et al. (2018). |
| Phonons Mode Peak | 1265 | Phonon mode peak refers to the peak in the phonon spectrum caused by specific phonon modes. Data from Petretto et al. (2018). |
| Poisson Ratio | 1181 | Poisson Ratio quantifies the ratio of transverse strain to axial strain in a material under uniaxial stress, reflecting its elastic deformation behavior. Data from De Jong et al. (2015a). |
| Poly Electronic | 1056 | The Average eigenvalue of the dielectric tensor's electronic component, where the dielectric tensor links a material's internal and external fields. Data from Petousis et al. (2017). |
| Poly Total | 1056 | The Average dielectric tensor eigenvalue. Data from Petousis et al. (2017). |
| Piezoelectric Modulus | 941 | Piezoelectric modulus measures a material's ability to convert mechanical stress into electric charge or vice versa. Data from De Jong et al. (2015b). |

adopt mean pooling of embedding for all properties since it performs significantly better than sum pooling in certain tasks (e.g. band gap prediction), which echos the findings in Shoghi et al. (2024).

For the Adaptive Module Composition (AMC) algorithm, we set the number of nearest neighbors ($K$ in Eq. (1)) to 5. For the optimization problem formulated in Eq. (3), we utilize the CPLEX optimizer from the cvxpy package (Diamond et al., 2014). AMC is applied separately for each random split of the downstream tasks to avoid data leakage.

**Computational Cost**   Experiments are conducted on NVIDIA A100 80 GB GPUs. During the module training stage, training time ranges from 30 to 300 GPU hours, depending on the dataset size. While this training process is computationally expensive, it is a one-time investment, as the trained models are stored in MoMa Hub as reusable material knowledge modules. Downstream fine-tuning requires significantly less compute, ranging from 2 to 8 GPU hours based on the dataset scale. The full module and adapter module require similar training time; however, the adapter module greatly reduces memory consumption during training. The time cost of AMC is discussed in Section D.3.

## C.3   BASELINE DETAILS

The CGCNN baseline refers to fine-tuning the CGCNN model (Xie & Grossman, 2018) separately on 17 downstream tasks. Conversely, MoE-(18) involves training individual CGCNN models for

each dataset in MoMa Hub and subsequently integrating these models using mixture-of-experts (Jacobs et al., 1991; Shazeer et al., 2016). For the baseline results of CGCNN and MoE-(18), we reproduce the results with the open-source codebase provided by Chang et al. (2022) and follow the exactly same hyper-parameters as reported in their papers.

For UMA, we fine-tune the UMA-Medium checkpoint (the largest open-sourced UMA model) on each dataset. To determine the batch size, we follow the max-atom batching strategy from the original UMA paper and set the maximum atoms per batch to 200, which ensures consistent memory usage across systems of varying sizes. All remaining hyperparameters (e.g., learning rate, number of epochs) follow the configurations used in JMP baselines.

For JMP-FT, we use the JMP (large) checkpoint from the codebase open-sourced by Shoghi et al. (2024) and fine-tune it directly on the downstream tasks with a batch size of 64. JMP-MT adopts a multi-task pre-training strategy, training on all 18 MoMa Hub source tasks without addressing the conflicts between disparate material tasks. Starting from the same pre-trained checkpoint as JMP-FT, JMP-MT employs proportional task sampling and trains for 5 epochs across all tasks with a batch size of 16. The convergence of multi-task pre-training is indicated by a lack of further decrease in validation error on most tasks after 5 epochs. For downstream fine-tuning, both JMP-FT and JMP-MT adopt the same training scheme as the fine-tuning stage in MoMa.

### C.4 Implementation Details of LoraHub Learning & Softmax Weighting

In our analysis experiments (Section 3.3), we compare AMC against two alternative module composition strategies: *LoraHub Learning*, a black-box optimization approach, and *Softmax Weighting*, a non-optimized performance-based heuristic.

For the implementation of LoraHub Learning, we strictly follow the hyper-parameters and black-box optimization scheme in its official repository except that we use a training-free $k$NN predictor to obtain the metric in each round of optimization, which is aligned with AMC. This is because current capabilities pre-trained models cannot enable zero-shot prediction of material tasks as in LLMs.

For the implementation of Softmax Weighting, we convert the predicted MAE from the same initial $k$NN evaluation into a weight for each module. The goal is to directly assign higher weights to modules with better predicted individual performance (i.e., lower MAE). Formally, the weight $w_j$ for module $j$ is calculated as:

$$w_j = \frac{\exp(-\text{MAE}_j/T)}{\sum_{k=1}^{N} \exp(-\text{MAE}_k/T)} \tag{4}$$

where the temperature $T$ is set to 1.

### C.5 Discussion for AMC analysis experiments

For the router-based JMP-(18) approach, full fine-tuning all parameters induces formidable memory cost, and is impractical considering MoMa Hub may further scale in the future. Hence, resembling Chang et al. (2022), we only unfreeze the final MLP layer as well as the router network in downstream fine-tuning. We believe it underperforms MoMa because training a router over 18 heterogeneous experts with limited supervision per task is intrinsically difficult, leading to unstable and suboptimal training of module selection. By contrast, AMC avoids router training and uses a training-free convex weighting scheme guided by kNN-based proxy error, which is much better suited to the data-scarce, highly disparate material setting.

We conjecture that AMC outperforms LoraHub Learning for two main reasons. First, LoraHub optimizes weights based on the composed module, where arbitrary mixtures of heterogeneous representations yield noisy error signals and a rugged, non-convex landscape. Second, AMC decouples weight selection from feature mixing by optimizing ensemble predictions. This formulation transforms the task into a convex problem, enabling AMC to reliably converge to a global optimum without navigating the instability inherent to search-based methods.

The advantage of AMC over the Softmax Weighting highlights the importance of optimizing for synergy. Softmax Weighting determines each module's contribution based solely on its isolated performance, overlooking potential synergistic interactions. In contrast, AMC explicitly optimizes for the weight configuration that maximizes collective performance and captures such interactions.

## C.6 Details on MoMa Hub Scaling Analysis

The QM9 dataset (Ramakrishnan et al., 2014) comprises 12 quantum chemical properties (including geometric, electronic, energetic, and thermodynamic properties) for 134,000 stable small organic molecules composed of CHONF atoms, drawn from the GDB-17 database (Ruddigkeit et al., 2012). It is widely served as a comprehensive benchmarking dataset for prediction methods of the structure-property relationships in small organic molecules.

In the continual learning experiments, we expand the MoMa hub by including modules trained on the QM9 dataset. For module training, we adopt the same training scheme as the original MoMa modules, with the exception of using sum pooling instead of mean pooling, as it has been empirically shown to perform better (Shoghi et al., 2024).

# D    More Experimental Results

## D.1 Correlation analysis between kNN-based proxy and post-fine-tuning performance

We empirically examine whether the kNN-based proxy error used in AMC is a reliable indicator of post–fine-tuning performance. We consider three representative targets—Refractive Index, Phonons Mode Peak, and Exfoliation Energy—covering optical, vibrational, and energetic material properties. For each task, we use all 18 modules in the MoMa hub and record (i) the per-module proxy MAE computed during the kNN step of AMC, and (ii) the test MAE obtained by fine-tuning each module individually on the same target. We then compute the Spearman and Pearson correlations between the proxy MAE and the post–fine-tuning MAE over the 18 modules. Across all three targets, we observe consistently strong positive correlations (Spearman $> 0.8$, Pearson $> 0.6$, with p-values $< 0.01$ for Pearson and $< 0.0001$ for Spearman). Concretely, the Pearson correlations are 0.603, 0.628, and 0.699 for Phonons Mode Peak, Refractive Index, and Exfoliation Energy, respectively; the corresponding Spearman correlations are 0.851, 0.825, and 0.816. As visualized in Fig. 5, modules that achieve lower proxy error systematically attain lower post–fine-tuning MAE, providing direct empirical support that the kNN-based proxy is a reliable signal for guiding weight optimization in AMC.

## D.2 Sensitivity Analysis of kNN Proxy and Optimizer

We perform an additional sensitivity analysis of the kNN components in AMC. Specifically, we vary the number of neighbors $k$, switch the distance metric from cosine similarity to MAE, and modify the normalization of similarity scores from a weighted average to a uniform average over the kNN set. We study robustness by (i) computing the average pairwise correlation of the resulting module weight vectors across variants, and (ii) comparing the final post–fine-tuning MAE across variants.

Table 6 shows that the learned composition weights are highly robust under different kNN configurations. Over the 17 datasets, the average pairwise Pearson and Spearman correlations of the weight vectors are typically above $0.7$. This indicates that changing $k$, the distance metric, or the normalization scheme perturbs the proxy predictor but AMC consistently recovers a very similar relative weighting over modules, i.e., the inferred relationships between modules remain stable.

Table 7 reports the resulting post–fine-tuning MAEs for each kNN variant. This analysis is conducted on a single train/validation split and a single random seed per dataset. Even under this stringent setting, MAE remains reasonably stable for most tasks across the five kNN configurations. The main exception is the *Elastic Anisotropy* dataset, where we observe larger variation between variants; in our main experiments we already noted substantial fluctuations across random seeds for this target. Elastic anisotropy is a derived mechanical metric that depends on the full elastic response of the material, and we find in practice that the corresponding mapping from structure to target is more challenging and sensitive to initialization, which can amplify small differences in the proxy into larger differences in final MAE.

Finally, we also varied the CPLEX optimizer tolerances (optimality and MIP gap) from the default $10^{-6}$ to $10^{-3}$ and $10^{-9}$. These changes had no effect on the optimized weights, which is consistent

| Dataset | Avg. Pearson Corr. of Weights | Avg. Spearman Corr. of Weights |
|---|---|---|
| Experimental Band Gap | 0.9776 | 0.7553 |
| Formation Enthalpy | 0.9972 | 0.8300 |
| 2D Dielectric Constant | 0.9752 | 0.7545 |
| 2D Formation Energy | 0.9935 | 0.7495 |
| Exfoliation Energy | 0.6139 | 0.8193 |
| 2D Band Gap | 0.9594 | 0.5308 |
| 3D Poly Electronic | 0.7660 | 0.7711 |
| 3D Band Gap | 0.9865 | 0.9267 |
| Refractive Index | 0.8591 | 0.8249 |
| Elastic Anisotropy | 0.9781 | 0.8910 |
| Electronic Dielectric Constant | 0.9761 | 0.7768 |
| Dielectric Constant | 0.9731 | 0.7486 |
| Phonons Mode Peak | 0.9349 | 0.8936 |
| Poisson Ratio | 0.8570 | 0.8511 |
| Poly Electronic | 0.8876 | 0.8222 |
| Poly Total | 0.9309 | 0.7178 |
| Piezoelectric Modulus | 0.7319 | 0.7972 |

Table 6: **Stability of AMC weights under different kNN variants.** Average pairwise Pearson and Spearman correlations between module weight vectors obtained from varying $k$, the distance metric, and the similarity normalization. Across most datasets, both correlations are typically above $0.7$, indicating that AMC recovers highly consistent relative weight patterns over modules despite changes in the kNN setup.

| Dataset | weighted_cos_K3 | weighted_cos_K5 | weighted_cos_K10 | weighted_mae_K5 | uniform_cos_K5 |
|---|---|---|---|---|---|
| Experimental Band Gap | 0.2139 | 0.2284 | 0.2439 | 0.2356 | 0.2314 |
| Formation Enthalpy | 0.0142 | 0.0156 | 0.0181 | 0.0123 | 0.0164 |
| 2D Dielectric Constant | 0.3152 | 0.3096 | 0.3137 | 0.3251 | 0.3120 |
| 2D Formation Energy | 0.0292 | 0.0339 | 0.0401 | 0.0261 | 0.0384 |
| Exfoliation Energy | 0.6078 | 0.6395 | 0.7421 | 0.7084 | 0.6753 |
| 2D Band Gap | 0.1529 | 0.1468 | 0.1574 | 0.1414 | 0.1485 |
| 3D Poly Electronic | 0.5243 | 0.5093 | 0.5138 | 0.5207 | 0.5038 |
| 3D Band Gap | 0.0262 | 0.0295 | 0.0301 | 0.0237 | 0.0303 |
| Refractive Index | 0.2503 | 0.2560 | 0.2624 | 0.2626 | 0.2576 |
| Elastic Anisotropy | 0.0811 | 0.1108 | 0.1856 | 0.3819 | 0.5717 |
| Electronic Dielectric Constant | 0.3564 | 0.3532 | 0.3585 | 0.3393 | 0.3560 |
| Dielectric Constant | 0.5257 | 0.5359 | 0.5595 | 0.5604 | 0.5442 |
| Phonons Mode Peak | 0.1062 | 0.1310 | 0.1869 | 0.1446 | 0.1414 |
| Poisson Ratio | 0.3116 | 0.3345 | 0.3802 | 0.3546 | 0.3455 |
| Poly Electronic | 0.7307 | 0.7684 | 0.8036 | 0.7265 | 0.7691 |
| Poly Total | 0.5606 | 0.5848 | 0.5969 | 0.5850 | 0.5837 |
| Piezoelectric Modulus | 0.5843 | 0.5940 | 0.6093 | 0.5971 | 0.6002 |

Table 7: **Sensitivity of post–fine-tuning MAE to kNN design choices.** Test MAE for different kNN configurations in AMC: varying $k$ (3/5/10), distance metric (cosine vs. MAE), and normalization (weighted vs. uniform) across 17 datasets. For most tasks, MAE differences between variants are modest, showing that downstream performance is relatively stable with respect to kNN design choices.

with our formulation: our optimization problem is a small and strongly convex MIQP over continuous variables, and the solver consistently reaches the global optimum under all tested settings.

### D.3 EFFICIENCY ANALYSIS OF AMC

**Time Cost** For the prediction estimation stage, we further divide it into the embedding generation and kNN prediction step. While these steps should be conducted separately for each module and each downstream dataset, the process can be parallelized and the runtime mainly depends on the size of the downstream dataset. As shown in Table 8, the maximum total time is below 30 seconds. For the weight optimization stage, we report the minimum and maximum time required for convergence of each downstream task (Eq. (3)). As shown in Table 9, the time cost is negligible and remains roughly constant as the number of modules scales.

Table 8: Module prediction estimation time

|  | Min time (s) | Max time (s) |
|---|---|---|
| Embedding generation | 7.29 | 24.06 |
| KNN prediction | 0.05 | 4.02 |
| Total time | 7.34 | 28.08 |

Table 9: Weight optimization time

| Module # | Min time (s) | Max time (s) |
|---|---|---|
| 3 | 0.07 | 0.08 |
| 9 | 0.12 | 0.15 |
| 18 | 0.14 | 0.25 |

Table 10: Test set MAE and average test loss of JMP-FT and MoMa under the full-data, 100-data, and 10-data settings. Results are averaged over five random data splits on one random seed. Results are preserved to the third significant digit.

| Datasets | JMP-FT | MoMa | JMP-FT (100) | MoMa (100) | JMP-FT (10) | MoMa (10) |
|---|---|---|---|---|---|---|
| Experimental Band Gap | 0.380 | 0.305 | 0.660 | 0.469 | 1.12 | 1.245 |
| Formation Enthalpy | 0.156 | 0.0821 | 0.273 | 0.101 | 0.514 | 0.143 |
| 2D Dielectric Constant | 2.45 | 1.90 | 3.19 | 2.35 | 7.74 | 3.31 |
| 2D Formation Energy | 0.135 | 0.0470 | 0.366 | 0.113 | 0.842 | 0.214 |
| 2D Exfoliation Energy | 38.9 | 36.1 | 54.4 | 56.1 | 118 | 87.3 |
| 2D Band Gap | 0.611 | 0.366 | 0.890 | 0.517 | 1.23 | 1.05 |
| 3D Poly Electronic | 23.7 | 23.0 | 33.6 | 24.8 | 54.0 | 48.9 |
| 3D Band Gap | 0.249 | 0.201 | 1.71 | 0.686 | 2.10 | 1.47 |
| Dielectric Constant | 0.0552 | 0.0535 | 0.134 | 0.102 | 0.289 | 0.231 |
| Elastic Anisotropy | 2.11 | 2.85 | 4.85 | 3.79 | 4.02 | 5.26 |
| Electronic Dielectric Constant | 0.108 | 0.0903 | 0.260 | 0.178 | 0.568 | 0.500 |
| Total Dielectric Constant | 0.172 | 0.155 | 0.361 | 0.287 | 0.543 | 0.527 |
| Phonons Mode Peak | 0.0710 | 0.0521 | 0.221 | 0.199 | 0.493 | 0.485 |
| Poisson Ratio | 0.0221 | 0.0203 | 0.0345 | 0.0317 | 0.0466 | 0.057 |
| Poly Electronic | 2.10 | 2.13 | 3.24 | 2.88 | 6.08 | 5.10 |
| Total Poly | 4.83 | 4.76 | 6.54 | 6.32 | 11.2 | 10.1 |
| Piezoelectric Modulus | 0.169 | 0.175 | 0.248 | 0.258 | 0.303 | 0.290 |
| **Average Normalized Test MAE** | 0.222 | 0.187 | 0.408 | 0.299 | 0.700 | 0.550 |

**Memory Cost**  During embedding generation, only one module is loaded into GPU at a time, requiring approximately 1.8 GB of memory. The generated embeddings are stored on CPU, with the largest set requiring about 5.5 MB. Overall, AMC is lightweight in memory usage and scales well with the number of modules.

### D.4  COMPLETE FEW-SHOT LEARNING RESULTS

We present the complete results of the few-shot learning experiments in Table 10. MoMa consistently shows performance improvements across all settings, with the margin of normalized test loss increasing as dataset size shrinks. These results highlight MoMa's strong potential to retain a performance advantage in few-shot scenarios, which are prevalent in material property prediction tasks.

### D.5  COMPLETE RESULTS FOR SCALING ANALYSIS OF MOMA

We present the complete results for the scaling analysis of MoMa in Table 11. We report test set MAE for hub sizes of 5, 10, 18 (full MoMa hub), and 30 modules (by adding QM9 modules). The last row reports the normalized average over all tasks.

## E  POTENTIAL SOCIETAL IMPACT

MoMa is visioned to be an open-source platform for the sharing of materials knowledge as modules. Potential positive societal impacts include the acceleration of the discovery of new materials with desirable properties, which benefit industries such as energy, electronics, and manufacturing. However, there are risks associated with the mal-intended use of material knowledge to develop harmful or unsafe materials. To mitigate these risks, it is crucial to ensure that the application of this work adheres to ethical guidelines. Although we do not foresee significant negative consequences in the near future, we recognize the importance of responsible usage and oversight in the application of these technologies.

Table 11: Test set MAE of MoMa under different hub sizes: 5 modules, 10 modules, full MoMa hub (18 modules), and 30 modules (with QM9). Results are preserved to the fourth decimal digit.

| Number of MoMa Modules | 5 | 10 | 18 (MoMa) | 30 (+QM9) |
|---|---|---|---|---|
| Experimental Band Gap | 0.3478 | 0.3324 | 0.2975 | 0.2960 |
| Formation Enthalpy | 0.0799 | 0.0814 | 0.0789 | 0.0819 |
| 2D Dielectric Constant | 2.2075 | 1.9482 | 1.9406 | 1.8879 |
| 2D Formation Energy | 0.0513 | 0.0510 | 0.0438 | 0.0470 |
| 2D Exfoliation Energy | 38.6231 | 36.6587 | 34.5769 | 35.1542 |
| 2D Band Gap | 0.4624 | 0.4256 | 0.3649 | 0.3605 |
| 3D Poly Electronic | 23.3909 | 23.0813 | 22.7205 | 23.3679 |
| 3D Band Gap | 0.3035 | 0.2555 | 0.2270 | 0.2053 |
| Dielectric Constant | 0.0549 | 0.0529 | 0.0511 | 0.0529 |
| Elastic Anisotropy | 1.9967 | 2.4103 | 2.5340 | 2.6408 |
| Electronic Dielectric Constant | 0.1046 | 0.0878 | 0.0909 | 0.0892 |
| Total Dielectric Constant | 0.1762 | 0.1554 | 0.1571 | 0.1561 |
| Phonons Mode Peak | 0.0528 | 0.0505 | 0.0512 | 0.0460 |
| Poisson Ratio | 0.0240 | 0.0207 | 0.0206 | 0.0206 |
| Poly Electronic | 2.0588 | 2.0215 | 2.0445 | 1.9837 |
| Total Poly | 4.9129 | 4.9148 | 4.8804 | 4.7358 |
| Piezoelectric Modulus | 0.1805 | 0.1713 | 0.1721 | 0.1743 |
| **Average Normalized Test MAE** | 0.2040 | 0.1910 | 0.1853 | 0.1759 |

