# OpenReview forum: "MoMa: A Simple Modular Learning Framework for Material Property Prediction"
_ICLR.cc/2026/Conference — ICLR 2026 Poster_

### Official Review · Reviewer_1mck · 2025-10-22

**Soundness:** 2
**Presentation:** 3
**Contribution:** 2
**Rating:** 4
**Confidence:** 3

**Summary:**

This paper introduces an approach for material property prediction that trains specialised modules for different tasks and then adaptively combines them for new prediction scenarios. An algorithm is proposed to efficiently estimate the performance of each module on a target task and then optimise a weighted combination of these modules to create a tailored model for the specific task.  Experiments across 17 datasets show the effectiveness of the approach, especially in few-shot settings, enabling continual learning and scalable integration.

**Strengths:**

1. The evaluation spans 17 diverse downstream tasks. The authors compare  against strong and relevant baselines, including not just older methods (CGCNN) but state-of-the-art pre-trained models fine-tuned in different ways (JMP-FT, JMP-MT).
2. The authors have diligently validated their design choices. The ablation study presented in Figure 4 is particularly effective, clearly demonstrating that both the selection of modules (vs. averaging all) and the learned weighting (vs. uniform averaging) are critical to AMC's success.

**Weaknesses:**

1. The method of weighted-averaging fine-tuned model parameters seems to be nearly identical to the 'Model Soups' approach [1]
2. The algorithm's effectiveness hinges on the assumption that the kNN-based proxy error (Eq. 2) is a reliable indicator of final, post-fine-tuning performance. This is a strong heuristic, but its validity is not fully demonstrated. The authors should conduct an experiment correlating the proxy error of individual modules with their actual performance after being fine-tuned alone on the downstream task.
3. Section 3.5 presents a 'Continual Learning' experiment by adding 12 QM9 molecular modules to the Hub. While this demonstrates the framework's ability to incorporate new knowledge, it doesn't align with the standard definition of continual learning, which typically involves sequential training and mitigating 'catastrophic forgetting.'


[1] Model soups: averaging weights of multiple fine-tuned models improves accuracy without increasing inference time, In ICML'22.

**Questions:**

1. Would inclusion of a “Naive Soup” baseline, averaging all eighteen fine-tuned modules with uniform weights, help isolate AMC’s benefits by demonstrating whether ensembling everything is inferior to selection and weighting?
2. Could validation be provided showing correlation between the kNN-based proxy error and post-fine-tuning performance, e.g., scatter plots comparing modules’ proxy errors versus their results after standalone fine-tuning across representative tasks?
3. Could sensitivity analyses for K in kNN, distance metric choices, normalization schemes, and optimizer tolerance be included, reporting how MAE and selected weights vary across these settings for robustness measures?
4. Could a second backbone be added to main results, even on a subset, to validate backbone-agnostic gains, and discuss layer alignment issues impacting weight-space merging beyond primarily JMP-based experiments reported?

---

> ### Author Response · Authors · 2025-11-24
>
> # Response 1/2
>
> We appreciate the reviewer for the constructive and insightful comments. We have addressed all your concerns in the following, and any further discussions are welcome.
>
> ## Q1: About the comparison with 'Model Soups'
>
> Thanks for the suggestion. Actually, we already include a naive soup baseline in **Section 4.3** as the *"All Average"* variant, where all 18 fine-tuned modules are merged with uniform weights. As shown in **Figure 6.b**, the "All Average" / "Naive Soup" method lags behind AMC on **15 out of 17** tasks, with an average test MAE increase of **18.0%**, which highlights the benefit of AMC’s learned weighting. To clarify this connection, we have added an explicit citation to Model Soup in our revised paper.
>
> ## Q2: About validating the use of the kNN-based proxy
>
> Thanks for pointing out the need to directly validate the kNN-based proxy as an indicator of post–fine-tuning performance. We now provide strong validation for our use of proxy through an explicit correlation analysis.
>
> Specifically, we conduct further analysis on three representative tasks: the prediction of *Refractive Index*, *Phonons Mode Peak* and *Exfoliation Energy,* which respectively represent the optical, vibrational and energetic properties of materials. For each task, we record the 18 per-module kNN MAEs computed during the kNN step of AMC, and the corresponding 18 post-finetuning MAEs with each standalone module. Then we calculate the Spearman and Pearson correlations between the kNN error and the post-fine-tuning error:
>
> | Task                 | Phonons Mode Peak | Refractive Index | Exfoliation Energy |
> | -------------------- | ----------------- | ---------------- | ------------------ |
> | Pearson correlation  | 0.6032            | 0.6281           | 0.6986             |
> | Spearman correlation | 0.8514            | 0.8246           | 0.8163             |
>
> Across all three tasks, we observe consistently strong correlations (Spearman **>0.8** and Pearson **>0.6**, with p-values **< 0.01** for Pearson and **< 0.0001** for Spearman), which provide strong empirical support that the kNN-based proxy is a reliable indicator of final performance and justifies its use to guide weight optimization in AMC. In our revised paper, we also include a scatter plots visualizing these relationships in **Fig. 5 of Section 3.2**.
>
> ## Q3: About the use of term "Continual Learning"
>
> We agree that our use of the term "continual learning" may induce misunderstandings, so we will change the title from "Continual Learning" to "Scaling Analysis of MoMa Hub Modules" and replace this phrase throughout the paper. **The revised Section 4.5** now include an in depth discussion of MoMa's behavior when the number of modules grows to **5, 10, 18 and 30**, highlighting MoMa's potential to further benefit from larger-scale hubs in realistic material discovery scenarios. We will post another comment after we finish this revision.
>
> ## Q4: About adding a second backbone
>
> Actually, we have already reported Orb-v2 results previously in Appendix D.1 which are now moved to **Section 4.3**. We now report the full average test MAEs: Orb-v2-MoMa outperforms Orb-v2-FT on 13 out of 17 tasks, with an average MAE reduction of **6.1%**. Note that the Orb-v2 model is based on a GNS architecture, which is non-equivariant and substantially simpler than the GemNet-based architecture used in JMP. This provides direct evidence that the benefits of the MoMa framework **generalize across different encoder families**, rather than being specific to a single architecture.
>
> |                                | Orb-v2-FT | Orb-v2-MoMa |
> | ------------------------------ | --------- | ----------- |
> | Experimental Band Gap          | 0.3735    | **0.3183**  |
> | Formation Enthalpy             | 0.1351    | **0.1009**  |
> | 2D Dielectric Constant         | 2.1667    | 2.3199      |
> | 2D Formation Energy            | 0.1187    | **0.0915**  |
> | Exfoliation Energy             | 35.1708   | **34.8223** |
> | 2D Band Gap                    | 0.5452    | **0.4817**  |
> | 3D Poly Electronic             | 24.8645   | **24.1720** |
> | 3D Band Gap                    | 0.4006    | **0.3153**  |
> | Refractive Index               | 0.0781    | **0.0742**  |
> | Elastic Anisotropy             | 3.0569    | 3.1776      |
> | Electronic Dielectric Constant | 0.1436    | **0.1314**  |
> | Dielectric Constant            | 0.2575    | **0.2363**  |
> | Phonons Mode Peak              | 0.0772    | 0.0907      |
> | Poisson Ratio                  | 0.0293    | **0.0273**  |
> | Poly Electronic                | 2.5573    | 2.6176      |
> | Poly Total                     | 5.7060    | **5.6500**  |
> | Piezoelectric Modulus          | 0.2257    | **0.2150**  |

---

> ### Author Response · Authors · 2025-11-24
>
> # Response 2/2
>
> ## Q5: About the sensitivity of kNN and optimizer
>
> As requested, we perform an in depth sensitivity analysis of the kNN components in AMC. Specifically, we vary the number of neighbors k, switch the distance metric from cosine similarity to MAE, and modify the normalization of similarity scores from a weighted average to a uniform average over the kNN set. We study robustness by computing the average pairwise correlation of the corresponding weight distributions and  comparing the final post-finetuning MAE across variants. Due to space limit, the result tables are shown in Table 6 and 7 of **Appendix D.2** in our revised paper.
>
> We first note that **the selected weights are highly robust across all kNN variants.** Over the 17 datasets, the average pairwise correlation of the weight vectors is typically **above 0.7**. This indicates that changing K, the distance metric, or the normalization scheme perturbs the proxy predictor but AMC consistently recovers a very similar relative weighting over modules, i.e., the inferred relationships between modules are stable.
>
> Regarding final post-finetuning MAE, this analysis is conducted on a single data split and a single random seed per dataset. Even under this setting, **MAE remains reasonably stable for most tasks**. However, we do notice the *Elastic Anisotropy* dataset exhibits a large variation. In our main experiments we already noted large fluctuations when using different random seeds for this dataset. It is a high-noise, derived mechanical property, whose mapping from structure to target is intrinsically harder to learn and more sensitive to different initializations.
>
> Finally, varying the CPLEX optimizer tolerances (optimality and MIP gap) from the default 1e-6 to 1e-3 or 1e-9 **has no effect on the optimized weights**, which is expected given that our optimization problem is a small and strongly convex MIQP problem in continuous variables, hence the optimizer could consistently reach the global optimum.
>
> ## Q6: About layer alignment in weight merging
>
> We agree with the reviewer that weight-space merging in general can be affected by layer alignment issues. In MoMa, we mitigate this in two ways.
>
> First, all MoMa modules are fine-tuned from the **same pretrained backbone checkpoint**. Recent work on cross-task linearity [1] shows that models initialized from a common pretrained checkpoint and fine-tuned on different tasks often lie in a approximately linear region of the feature space, so weight interpolation tends to remains in a good basin.
>
> Second, unlike naive uniform weight averaging, **AMC** uses a data-driven optimization over module weights based on a proxy error; modules that are poorly aligned or interfere negatively with the target task are assigned negligible weights and effectively excluded from the final merge. This makes MoMa less sensitive to misaligned modules than uniform soups or indiscriminate averaging.
>
> [1] On the Emergence of Cross-Task Linearity in the Pretraining-Finetuning Paradigm, ICML 2024

---

### Official Review · Reviewer_3v4v · 2025-11-01

**Soundness:** 3
**Presentation:** 3
**Contribution:** 3
**Rating:** 4
**Confidence:** 2

**Summary:**

This paper introduces MoMa, a simple modular learning framework to overcome the diversity and disparity challenges in material property prediction. It first trains specialized modules on a wide range of tasks and stores them in a MoMa Hub. Then, Adaptive Module Composition (AMC) composes the most synergistic modules from this Hub into a single, tailored model for new tasks, achieving substantial performance gains over existing baselines.

**Strengths:**

1. This paper effectively tackles the challenge of material learning by applying a modular deep learning framework.

2. The paper is clearly written, presenting an intuitive and well-motivated method.

3. MoMa achieves superior performance, showing substantial average improvements over existing baselines.

**Weaknesses:**

1. AMC's design relies heavily on heuristics, such as using a kNN performance proxy, and the paper would be stronger if it provided sufficient justification for this design choice.

2. As the authors note for future work, it is questionable whether the kNN-based proxy will remain sufficiently efficient and effective as MoMa scales to hundreds or thousands of modules.

**Questions:**

Please refer to the Weaknesses section above.

---

> ### Author Response · Authors · 2025-11-24
>
> We appreciate the reviewer for the constructive and insightful comments. We have addressed all your concerns in the following, and any further discussions are welcome.
>
> ## Q1: About the justification for using a kNN-based proxy
>
> We thank the reviewer for pointing out the need to further justify the design of AMC, in particular the kNN-based performance proxy. **We now provide both empirical and theoretical justification for this choice.**
>
> First, we add an empirical validation showing that a *lower* optimized proxy error in AMC is strongly correlated with *better* final post–fine-tuning performance across downstream tasks. On three representative tasks: Refractive Index (optical property), Phonons Mode Peak (vibrational property) and Exfoliation Energy (energetic property), we record the 18 proxy MAEs for each module calculated during the kNN step of AMC, and the 18 post-finetuning MAEs with each standalone module. Then we calculate the Spearman and Pearson correlation between proxy error and post-fine-tuning performance:
>
> | Task                 | mp_phonons | mp_dielectric | jarvis_2d_exfoliation |
> | -------------------- | ---------- | ------------- | --------------------- |
> | Pearson correlation  | 0.6032     | 0.6281        | 0.6986                |
> | Spearman correlation | 0.8514     | 0.8246        | 0.8163                |
>
> We believe these high correlations (Spearman **>0.8** and Pearson **>0.6**, with p-values **< 0.01** for Pearson and **< 0.0001** for Spearman) strongly validates our use of the kNN-based proxy metric to guide weight optimization.
>
> Second, we complement these results with a formal analysis showing that, for any weight vector w, the fine-tuning risk of the merged model is upper-bounded by the proxy risk up to controlled terms: $ R_{\mathrm{FT}}(w) \leq R_{\mathrm{proxy}}(w) + C\delta + \epsilon_1(n;w) + \varepsilon_{\mathrm{opt}}.$
>
> This bound makes explicit that minimizing the kNN-based proxy provides a principled surrogate for minimizing the final generalization error, rather than a purely heuristic criterion. Refer to Appendix B for the full analysis.
>
> ## Q2: About the efficiency and effectiveness of kNN-based proxy as MoMa further scales
>
> We thank the reviewer for raising this concern. We now show evidence that MoMa will remain efficient and effective even in larger scale.
>
> 1. **Efficiency.**
>
> AMC was designed explicitly to avoid the combinatorial explosion of evaluating all module subsets. The kNN–based proxy only requires running label propagation once per module on the target dataset and then solving a convex optimization problem in the weight space. As a result, its time and memory cost scale **linearly** with both the number of modules in the hub and the number of target examples, rather than exponentially in the number of module combinations. This linear structure allows for parallel computing techniques for acceleration.
>
> In **Appendix D.3** we report the wall-clock time and memory footprint of AMC: even on our largest datasets and with the largest hub, the AMC stage adds only a modest overhead (~30 seconds and ~5MB of CPU memory) compared to standard per-task fine-tuning.
>
> 2. **Effectiveness.**
>
> We have provided a further analysis of MoMa's effectiveness at scale , where we vary the hub size from **5 to 10, 18, and 30** modules. As shown in the table below, we observe a clear monotonic decrease in average normalized test set MAE of 17 downstream tasks as the hub grows, with no sign of saturation at 30 modules, indicating that the kNN-based proxy remains informative and beneficial when the hub becomes even larger. These results have been updated in **Section 4.5** of our revised paper.
>
> |                             | 5-modules | 10-modules | MoMa (18 modules) | 30-modules (QM9) |
> | --------------------------- | --------- | ---------- | ----------------- | ---------------- |
> | Average Normalized MAE | 0.2040    | 0.1910     | 0.1853            | 0.1759           |

---

### Official Review · Reviewer_nxqB · 2025-11-01

**Soundness:** 3
**Presentation:** 3
**Contribution:** 3
**Rating:** 8
**Confidence:** 4

**Summary:**

This paper presents a straightforward machine learning framework for materials science. The framework is composed of multiple independently trained modules, each tailored for a specific task. For a given downstream application, the framework dynamically selects and integrates a combination of modules based on their estimated predictive performance, followed by joint fine-tuning. Extensive experiments across 17 material property prediction benchmarks demonstrate the framework’s superior accuracy and adaptability.

**Strengths:**

The MoMa framework have individual modules trained on a wide range of properties and tasks. This architecture aligns with the diversity and disparity of material properties. It also enables the framework to extend to other modules trained on specialized material properties easily. By having a full module and an adapter module, the framework is parameter-efficient. It also allows easy fine-tuning on downstream tasks. The author also shows that the module selected by the algorithm correlates with the physical relationships between material properties, showing the effectiveness and trustworthiness of the algorithm.

**Weaknesses:**

The MoMa is a flexible framework for material property prediction. However, there might be other material properties that could not be captured by the trained modules, resulting in inaccurate property prediction.

**Questions:**

1. Does using the open source cvxpy package guarantee to find the optimal weight?
2. How much does the accuracy of individual modules affect the weight calculation and model performance?
3. Is there any correlation between individual modules? Does it affect molecule weight calculation or model performance?

---

> ### Author Response · Authors · 2025-11-25
>
> We appreciate the reviewer for the constructive and insightful comments. We have addressed all your concerns in the following, and any further discussions are welcome.
>
> ## Q1: There may be other material properties not captured in the trained modules.
>
> We agree that there will inevitably be properties that are only weakly represented in the current hub. However, our 17-task evaluation (covering optical, vibrational, energetic, dielectric, and mechanical properties) already captures a broad range of useful material behavior. Moreover, MoMa is designed as an **open, extensible** hub: new modules trained on additional properties can be flexibly incorporated into MoMa. Thus, although the current coverage of properties may be limited, extending MoMa to cover more diverse properties is an exciting future direction that is in line with our vision of MoMa as an open platform for modularizing and distributing materials knowledge.
>
> ## Q2: Does using cvxpy guarantee to find the optimal weight?
>
> **Yes. For the proxy objective defined in AMC, our cvxpy-based solver is guaranteed to find the optimal weights.** The weight optimization problem in AMC is formulated as a small convex quadratic program over the probability simplex; the objective is strongly convex and the constraints are convex, so the corresponding problem has a unique global optimum for the proxy error. In practice, we empirically observe that CVXPY indeed converges to this optimum and is very stable: varying the CPLEX optimality and MIP gap tolerances from $10^{-6}$ to $10^{-3}$ or $10^{-9}$ does not change the optimized weights or the achieved objective value.
>
> ## Q3: How does the accuracy of individual modules affect weight calculation and model performance?
>
> In general, **more accurate modules are assigned larger weights and have a greater influence on the final model's performance**. In AMC, weights are obtained by minimizing a kNN-based proxy error, so modules with lower proxy MAE naturally contribute more to reducing the objective. Empirically, across all 17 downstream tasks we compute the Spearman correlation between per-module proxy MAE and the learned AMC weights and observe **an average correlation of -0.50**, indicating that modules with lower proxy error tend to receive larger weights while higher-error modules are downweighted. This provides direct evidence that more accurate modules are more emphasized by AMC, whereas less accurate modules receive small weights and thus have a relatively minor impact on the final performance.
>
> ## Q4: Is there any correlation between individual modules? Does it affect module weight calculation or model performance?
>
> There are correlations between MoMa modules both statistically and physically. Statistically, we analyze correlations between modules at the level of their per-module performance profiles, measured by kNN proxy error across all 17 downstream tasks. Among all 153 module pairs, 74 pairs have a Spearman correlation **above 0.8**. Physically, the MoMa hub includes modules trained on the same target from different databases (e.g., formation energy from Materials Project and JARVIS) as well as closely related properties (e.g., dielectric constant and band gap).
>
> However, AMC remains effective even when modules are correlated. For the 74 pairs of highly correlated modules, the average correlation of their learned AMC weights is **only 0.08**. We observe that, even when two modules behave very similarly, AMC does not force them to share large or identical weights and tends to concentrate weight on the most informative ones, thereby reducing redundancy. Hence module correlation has limited impact on module weight calculation and overall MoMa performance.

---

### Official Review · Reviewer_tuSk · 2025-11-02

**Soundness:** 3
**Presentation:** 3
**Contribution:** 2
**Rating:** 4
**Confidence:** 4

**Summary:**

This paper proposes MoMa, a modular learning framework for material property prediction designed to tackle the diversity (heterogeneity of tasks and systems) and disparity (incompatible physical laws) in materials datasets.
It first trains a collection of task-specific modules (either full fine-tuned models or lightweight adapters) on high-resource material datasets, and later composes these modules adaptively for new downstream tasks via the proposed AMC algorithm, which estimates module relevance using kNN-based proxy prediction, optimizes non-negative weights through convex programming, merges module parameters via weighted averaging, and fine-tunes the composed module.

**Strengths:**

- Introducing a hub-and-composition paradigm is new to this domain and nicely parallels modular LLM work (AdapterHub, LoRAHub).
- AMC is simple, requiring only kNN embedding estimation and a small convex optimization step.
- Evaluation is pretty comprehensive.
- The visualization of module weights (Fig. 6) provides interesting physical correlations and hints at cross-property transferability.
- Open-sourcing MoMa Hub could foster standardized module reuse and privacy-aware data sharing in materials science.

**Weaknesses:**

- While AMC is well-engineered, its core ideas (kNN prediction + convex weighting + parameter averaging) largely repackage known components from model-merging and ensemble literature. The conceptual advance over prior modular or LoRAHub methods is modest.
- The paper lacks formal justification or analysis of why the convex proxy optimization correlates with post-fine-tuning accuracy, beyond empirical correlation (r = 0.69). A clearer theoretical connection between proxy error and final generalization would strengthen the contribution.
- Baselines such as JMP-MT and MoE-(18) are implemented in-house; comparisons to other modern material foundation models (e.g., MatterSim [1], Orb-v3 [2], UMA [3]) are missing, limiting claims of SOTA performance.
- Also, ablations explore only AMC variants, not architectural or hub-scale sensitivities.
- Although the framework is claimed backbone-agnostic, all results use JMP. It remains uncertain whether MoMa generalizes to other encoders or representations (e.g., E3NN [4], GemNet-OC [5]).
- While clear overall, the paper is somewhat long and repetitive; core algorithmic details could be condensed, and figures (e.g., Fig. 3) are descriptive but not deeply analytical. The continual-learning section feels exploratory and under-developed.
- The work focuses on crystalline and small-molecule systems; polymeric, amorphous, and multi-phase materials are not addressed, limiting generality.

[1] Yang, H., et al. "A Deep Learning Atomistic Model Across Elements, Temperatures and Pressures." arXiv preprint arXiv:2405.04967 (2024).

[2] Rhodes, Benjamin, et al. "Orb-v3: atomistic simulation at scale." arXiv preprint arXiv:2504.06231 (2025).

[3] Wood, Brandon M., et al. "UMA: A Family of Universal Models for Atoms." arXiv preprint arXiv:2506.23971 (2025).

[4] Geiger, Mario, and Tess Smidt. "e3nn: Euclidean neural networks." arXiv preprint arXiv:2207.09453 (2022).

[5] Gasteiger, Johannes, et al. "GemNet-OC: developing graph neural networks for large and diverse molecular simulation datasets." arXiv preprint arXiv:2204.02782 (2022).

**Questions:**

See weaknesses.

---

> ### Author Response · Authors · 2025-11-24
>
> # Response 1/3
> We appreciate the reviewer for the constructive and insightful comments. We have addressed all your concerns in the following, and any further discussions are welcome.
>
> ## Q1: About AMC's conceptual advance over prior modular methods
>
> While AMC indeed adopts standard components (e.g., kNN, convex optimization and parameter merging), its key conceptual advance lies in its **representation-driven, training-free pipeline for optimizing composition weights.** AMC is a principled, robust, and efficient pipeline designed for material property prediction settings with highly diverse and disparate upstream modules and data-scarce downstream tasks.
>
> Existing weighted-composition approaches generally fall into two categories, **both relying on downstream prediction error after module composition**, which fails to tackle the challenges in material settings:
>
> 1. **Search-based methods**, which first merge modules using a candidate weight vector and **iteratively** optimize these weights based on the downstream predictive performance of the composed model (e.g., LoRAHub). However, material tasks differ widely in both input structure (2D/3D crystals, molecules) and target semantics (energies, band gaps, elastic tensors, etc.). As a result, the modules in the MoMa Hub induce **highly disparate representation spaces**, and arbitrary mixtures tend to produce representations that are poorly aligned with the target task, yielding highly unstable or uninformative predictive signals. We also show empirical benefits of AMC with a direct comparison with LoraHub in **Section 4.4**.
> 2. **Router-based methods**, which train an additional router or fusion network to weight and combine modules (e.g. mixture-of-experts). In our setting, the large number and high heterogeneity of upstream modules, combined with limited supervision on downstream tasks, makes router training difficult. Empirical comparisons with MoE-(18) in the main table and JMP-(18) in **Section 4.4** confirm this limitation. Also, there is a scalibility challenge as we scale MoMa to dozens (and potentially hundreds) of modules.
>
> The representation-driven and training-free nature of AMC allows it to successfully tackle the challenges rooted in material settings:
>
> 1. **Representation-level probing instead of prediction-level search.** For each downstream task, AMC probes the latent representation geometry induced by each module via kNN label propagation, and uses this as a proxy to estimate module performance without ever forming arbitrary mixtures. This offers a white-box objective for weight selection rather than relying on noisy prediction-level signals.
> 2. **Training-free weight optimization.** Given these proxy scores, AMC solves a sparse convex optimization problem to obtain non-negative composition weights. This avoids training any additional parameters for weight allocation, naturally sidestepping both the data-scarcity and scalability issues that plague router-based approaches when the number of modules grows.

---

> ### Author Response · Authors · 2025-11-24
>
> # Response 2/3
> ## Q2: About the formal analysis between proxy error and final generalization
>
> We thank the reviewer for highlighting the need to formally justify why our convex proxy objective is aligned with post–fine-tuning performance. In the revised version, we provide a formal analysis of the risk under the data distribution, showing that **the proxy risk controls the downstream fine-tuning risk of the merged model**. Specifically, we establish that for any weight vector w: $R_{\mathrm{FT}}(w) \leq R_{\mathrm{proxy}}(w) + C\delta + \epsilon_1(n;w) + \varepsilon_{\mathrm{opt}}.$
>
> In words, this inequality states that the fine-tuning risk $R_{\mathrm{FT}}(w)$ of the merged model **cannot exceed the proxy risk $R_{\mathrm{proxy}}(w)$ by more than a bounded margin**. This margin is fully characterized by: (i) the smoothness of the underlying regression function (absorbed into the constant $C$), (ii) a mild representation-closeness term $\delta$ inspired by empirical findings on Linear Mode Connectivity, (iii) the kNN estimation error $\epsilon_1(n;w)$, and (iv) the fine-tuning optimization error $\varepsilon_{\mathrm{opt}}$.
>
> Thus, whenever a choice of weights achieves low proxy risk, the corresponding merged-and-fine-tuned model is guaranteed to have low fine-tuning risk up to these explicitly controlled terms. This provides a clear theoretical connection between the convex proxy objective and the final generalization performance, and shows that our method is **not merely heuristic but optimizes a principled surrogate** of downstream risk. We include the full theorem and analysis in **Appendix B**.
>
> ## Q3: About the comparision with modern material foundation model
>
> We thank the reviewer for the suggestion and have added UMA as a new baseline. We fine-tune the UMA-M checkpoint (the largest open-sourced model for UMA) on each dataset and report the average test set MAE on 5 splits. Experiments are repeated on 5 random seeds.
>
> Note that we already include the results of Orb-v2 in Supplementary D.2 of our original paper, and we have updated this comparison in **Section 4.3**  the main text of our revised paper. Below we compare MoMa with Orb-v2 and UMA.
>
> | Experimental Band Gap          | Orb-v2      | UMA     | MoMa        |
> | ------------------------------ | ----------- | ------- | ----------- |
> | Formation Enthalpy             | 0.3735      | 0.3552  | **0.3052**  |
> | 2D Dielectric Constant         | 0.1351      | 0.1922  | **0.0821**  |
> | 2D Formation Energy            | 2.1667      | 2.3378  | **1.8947**  |
> | Exfoliation Energy             | 0.1187      | 0.1205  | **0.0471**  |
> | 2D Band Gap                    | **35.1708** | 44.3960 | 36.0535     |
> | 3D Poly Electronic             | 0.5452      | 0.4938  | **0.3655**  |
> | 3D Band Gap                    | 24.8645     | 32.6512 | **23.0366** |
> | Refractive Index               | 0.4006      | 0.2682  | **0.2008**  |
> | Elastic Anisotropy             | 0.0781      | 0.0582  | **0.0535**  |
> | Electronic Dielectric Constant | 3.0569      | 3.7854  | **2.8466**  |
> | Dielectric Constant            | 0.1436      | 0.1164  | **0.0903**  |
> | Phonons Mode Peak              | 0.2575      | 0.1833  | **0.1553**  |
> | Poisson Ratio                  | 0.0772      | 0.0811  | **0.0521**  |
> | Poly Electronic                | 0.0293      | 0.0225  | **0.0203**  |
> | Poly Total                     | 2.5573      | 2.3319  | **2.1295**  |
> | Piezoelectric Modulus          | 5.7060      | 5.6133  | **4.7564**  |
> | Piezoelectric Modulus          | 0.2257      | 0.2077  | **0.1751**  |
>
> **MoMa outperforms Orb-v2 and UMA in 16/17 and 17/17 tasks, with an average MAE decrease of 26.6% and 23.9%, which shows the strong empirical performance of MoMa.**
>
> Due to resource constraints, we currently do not compare Orb-v3 and MatterSim. However, there is evidence[1] that UMA-M outperforms Orb-v3 on canonical benchmarks (e.g., Matbench Discovery), so including UMA-M already provides a strong comparison point against modern material foundation models. We have updated the corresponding UMA results and analysis to **Section 4.2** of our main text.
>
> [1]: UMA: A Family of Universal Models for Atoms, Arxiv 2025

---

> ### Author Response · Authors · 2025-11-24
>
> # Response 3/3
> ## Q4: About more ablation on architectures or hub-scale sensitivities
>
> We added a hub-scale sensitivity experiment showing that larger module hubs improve overall performance. Starting from five randomly selected modules, we incrementally expanded the hub to ten, then to the full 18-module MoMa Hub, and finally to 30 modules with QM9. This setup simulates the hub's gradual expansion. Results are shown below.
>
> |                             | 5-modules | 10-modules | MoMa (18 modules) | 30-modules (QM9) |
> |-----------------------------|-----------|------------|-------------------|------------------|
> | Average normalized MAE | 0.2040    | 0.1910     | 0.1853            | 0.1759           |
>
> We can see with the scaling of MoMa hub, the average normalized test MAE of 17 tasks **consistently decreases**. We hypothesize that a larger and more diverse module hub offers a richer knowledge source and increase the likelihood that AMC can identify synergistic subsets tailored to the target downstream tasks. These results and analysis have been updated into **Section 3.5** of our revised paper.
>
> Since MoMa focuses on training paradigms instead of architectural modifications, we do not include architectural ablations.
>
> ## Q5: About the generalization to other encoders
>
> Actually, we have already reported Orb-v2 results in Appendix D.1. We now report the full average test MAEs below.
>
> |                                | Orb-v2-FT | Orb-v2-MoMa |
> | ------------------------------ | --------- | ----------- |
> | Experimental Band Gap          | 0.3735    | **0.3183**  |
> | Formation Enthalpy             | 0.1351    | **0.1009**  |
> | 2D Dielectric Constant         | 2.1667    | 2.3199      |
> | 2D Formation Energy            | 0.1187    | **0.0915**  |
> | Exfoliation Energy             | 35.1708   | **34.8223** |
> | 2D Band Gap                    | 0.5452    | **0.4817**  |
> | 3D Poly Electronic             | 24.8645   | **24.1720** |
> | 3D Band Gap                    | 0.4006    | **0.3153**  |
> | Refractive Index               | 0.0781    | **0.0742**  |
> | Elastic Anisotropy             | 3.0569    | 3.1776      |
> | Electronic Dielectric Constant | 0.1436    | **0.1314**  |
> | Dielectric Constant            | 0.2575    | **0.2363**  |
> | Phonons Mode Peak              | 0.0772    | 0.0907      |
> | Poisson Ratio                  | 0.0293    | **0.0273**  |
> | Poly Electronic                | 2.5573    | 2.6176      |
> | Poly Total                     | 5.7060    | **5.6500**  |
> | Piezoelectric Modulus          | 0.2257    | **0.2150**  |
>
> Orb-v2-MoMa outperforms Orb-v2-FT on 13 out of 17 tasks, with an average MAE reduction of **6.1%**. Note that the Orb-v2 model is based on a GNS architecture, which is non-equivariant and substantially simpler than the GemNet-based architecture used in JMP. This provides direct evidence that the benefits of the MoMa framework generalize across encoder families, rather than being specific to a single architecture.
>
> ## Q6: About the presentation
>
> We appreciate the reviewer’s feedback on our paper presentation. We have revised the manuscript to reduce redundancy in the algorithmic description of AMC in **Sec 3.2**, added a more analytical figure of AMC in **Fig.4**, and replaced the continual learning section with a principled scaling analysis of the MoMa Hub in **Sec. 4.6**. All major changes are **highlighted in blue**.
>
> ## Q7: About the generalization to other materials
>
> We acknowledge that the current work focuses primarily on crystalline and small organic-molecule systems, as discussed in our Limitations section. Nevertheless, these categories already encompass a wide spectrum of functional materials, including catalysts, OLED materials, and many others. We highlight the challenges for other classes of materials:
>
> - **Polymeric materials:** High-quality and standardized polymer datasets are still scarce, and their chain-like structures span multiple length scales, which current structure encoders cannot represent reliably. [1]
> - **Amorphous materials:** The absence of long-range order makes amorphous systems difficult to model, and existing ML models remain immature in capturing their complex local structural motifs. [2]
> - **Multi-phase materials:** These systems contain multiple interacting phases with different chemistries and structural behaviors, which leads to far higher complexity than single-phase crystalline materials and exceeds the capabilities of current encoders.
>
> Given these challenges, training MoMa modules for these material classes is currently less reliable and therefore not prioritized. However, we definitely expect to extend MoMa to these broader material families in the future.
>
> [1]: Long, Teng, et al. "Recent Progress of Artificial Intelligence Application in Polymer Materials." *Polymers,* 2025
>
> [2]: Zhang, Like, et al. "Application of Machine Learning in Amorphous Alloys." *Materials,* 2025

---

### Author Response · Authors · 2025-12-03
**Summary of Rebuttal and Manuscript Revision**

Dear Area Chair,

Thank you for assessing our paper during this unconventional phase of review; given the unusual circumstances of the venue, we sincerely appreciate your time and careful consideration. Below we provide a condensed summary of our responses for your assessment, demonstrating how our revisions have comprehensively addressed all reviewers' concerns. For convenience, all major changes in the updated manuscript are **highlighted in blue**.

During the rebuttal period, we are pleased that our work is recognized for:

- Introducing a **new-to-the-domain** hub-and-composition paradigm (Reviewer tuSk) that effectively tackles the **inherent diversity and disparity** of material properties through a modular framework; (Reviewers nxqB, 3v4v)
- Proposing the **simple and trustworthy** AMC algorithm that yields module weights **interestingly correlated** with physical relationships; (Reviewers tuSk, nxqB)
- Demonstrating **superior** performance over existing baselines (Reviewers 3v4v, 1mck), supported by **comprehensive** evaluations and **diligent** ablation studies. (Reviewers tuSk, 1mck)

While the unforeseen event has prevented our paper from engaging in any discussion with the reviewers, we are confident that we have addressed all their concerns by providing detailed response and manuscript revision.
Our revision mainly focuses on three core aspects:

**1. Rigorous justification for using the proxy error in AMC.**

Several reviewers (tuSk, 3v4v, 1mck) question the validity of using a kNN-based proxy error for weight optimization. We addressed this with **both** theoretical proofs and empirical verification:

- **Theoretical justification for proxy error**: We establish a formal error bound analysis in **Appx. B**, proving that minimizing the proxy risk controls the upper bound of the downstream fine-tuning risk. This confirms AMC optimizes a principled surrogate rather than a heuristic metric. (Reviewer tuSk)
- **Empirical correlations between proxy MAE and post-fine-tuning MAE**: We conduct a correlation study on three representative tasks (optical, vibrational, energetic). The results, now included in the main text (**Fig. 5; Appx. D.1**), reveal a strong positive correlation (Pearson **> 0.6**, Spearman **> 0.8**) between per-module proxy MAE and post-fine-tuning MAE, validating the proxy as a reliable predictor of final performance. (Reviewer 3v4v, 1mck)

**2. Enriched experiments to show the empirical strength of MoMa.**

Reviewers also raise valid empirical questions. In response, we have greatly enriched our experiments to show the generalizability, scalability and robustness of MoMa.

- **Comparison with modern material foundation models:** We compare MoMa against the state-of-the-art UMA model (added to **Tab. 1**). MoMa outperforms UMA on **all 17** tasks, achieving a substantial **23.9%** average MAE reduction. (Reviewer tuSk)
- **Generalization to New Architectures:** We extend our evaluation to the GNS-based backbone in **Sec. 4.3**. MoMa consistently outperforms direct fine-tuning with a **6.1%** average improvement, proving the framework is backbone-agnostic. (Reviewer tuSk, 1mck)
- **Hub-Scale & Sensitivity Analysis:** We add a scaling analysis of MoMa Hub in **Sec. 4.6**, showing a monotonous drop in normalized MAE (from **0.2040 to 0.1759**) as it scales from **5 to 30** modules. (Reviewer 3v4v) Moreover, sensitivity tests in **Appx. D.2** confirm that AMC generates highly consistent weights regardless of hyperparameter variations. (Reviewer 1mck, nxqB)

**3. Enhanced conceptual clarity and presentation.**

We have refined the manuscript to sharpen the conceptual advancement of AMC and improve readability, responding to Reviewer tuSk and 1mck.

- **Conceptual advance of AMC:** We explicitly distinguish AMC's *representation-driven, training-free pipeline* from prior paradigms (search-based black-box search methods (e.g. LoRAHub) and router-based methods (e.g. MoE)) and highlight the distinctive suitability of AMC to handle *high task disparity and data scarcity* in material settings. We add a review of relevant methods in **Sec. 2.2** and more carefully motivate AMC in **Sec. 3.2**, with empirical evidence highlighted in **Sec. 4.4**. (Reviewer tusk)
- **Refined presentation:** We rename "Continual Learning" to "Scaling Analysis of MoMa Hub" (**Sec 4.6**) and add explicit citations to Model Soup (**Sec. 4.3**) to prevent ambiguity (Reviewer 1mck). Also, we condense the algorithmic details in **Sec. 3.2** and provide a new analytical figure of AMC **(Fig. 4)**. (Reviewer tusk)

In summary, we include new theoretical analysis, empirical evidence, and conceptual clarifications that directly resolves all raised concerns. After the rebuttal, our manuscript have been substantially improved with enriched and more validated contribution.

We sincerely thank the AC for the careful evaluation, and hope our efforts affirm the value of our work to the ICLR community.

Best regards,

Authors

---

### Meta-Review · Area_Chair_pM6g · 2026-01-05

**Summary:**

The paper presents a new method for materials property prediction leveraging a collection of modular prediction models that can be applied to diverse cases of materials properties and materials systems, such as crystal structures and molecules. The MoMa framework works in two stages with the first being training or fine-tuning (full or with adapters) materials property models followed by Adaptive Module Composition (AMC) where the trained modules are deployed to solve a particular task. The paper presents a set of trained models in the MoMa hub for modeling a set of diverse materials properties and demonstrate general capabilities. The main novelty from the paper is the AMC-based routing method that uses kNN based clustering in the representation space of the hub models to minimize an ensemble proxy error of the composition of modules for determining the module composition for a particular materials property prediction query. The experimental results that MoMa presents competitive, and sometimes state-of-the-art results, for diverse property prediction tasks. Following the rebuttal, the paper also show MoMa achieving similarly competitive results with different backbone models. The paper also includes a set of ablations and analysis of the AMC method addressing effectiveness, scaling and correlation of AMC weights.

The reviewers generally praise the applicability of the method, including the extensible design approach of the MoMa framework, as well the extensive experimental testing. Some of the reviewers also positively highlight the intentional design choices of the framework and how the authors designed their analysis to justify their design choices.

The main concerns expressed by the reviewers relate to using heuristics-based justification for the AMC module and testing the module only on a single backbone. Similarly reviewers mentioned that having theoretical justifications for AMC would make the better stronger. The additional context and results provided by the authors during the discussion period addressed many of those concerns leading to edits incorporated into the paper. The edits also included further experiments and analysis of the scaling properties of MoMa in addition to further details on why the representation driven choice of AMC is appropriate for the materials space. Some reviewers also expressed concerns about generalizability for which the authors also provided additional clarifications and results.

The changes made by the authors during the discussion period made the paper significantly stronger while staying within the general framework of the initial set of core contributions.

**Reviewer Concerns:**

Addressed Concerns:
* Reviewer tuSk's concerns on theoretical justification and additional results with MoMa on diverse backbone models and additional ablations.
* Reviewer nxqB's clarifying questions.
* Reviewer 3v4v's concerns on justification for kNN proxy and a scaling analysis.
* Reviewer 1mck's concerns on proxy validation, using different model backbones and clarifications of the differentiation of AMC to prior methods.

Outstanding Concerns:
* Reviewer tuSk's concerns on novelty and differentiation from other techniques in hub-based modeling are partially addressed.
* Reviewer nxqB's generalization questions that can be deferred to future work.
* Reviewer 1mck's clarifications and additions could be better presented in the edited paper.

**Reviewer Scores:**

* Reviewer tuSk remains at 4.
* Reviewer nxqB remains at 8
* Reviewer 3v4v increases the score to 6.
* Reviewer 1mck increases the score to 6.

---

### Decision · Program_Chairs · 2026-01-26

Accept (Poster)